# TOWARDS STABLE LEARNING IN PREDICTIVE CODING NETWORKS

## ABSTRACT

Predictive coding (PC) offers a biologically plausible model of cortical functions, encompassing processes such as learning, prediction, encoding, and memory. However, predictive coding networks (PCNs) face significant challenges in stability and scalability, which constrain our capacity to elucidate cortical computation. Our study identifies instability in PCNs as a fundamental issue, focusing on the exponential growth of latent state norms and prediction errors after inference. These dynamics lead to exploding and vanishing gradients in PCNs. Moreover, the concentration of prediction errors near the input and output layer impedes effective learning, exacerbating performance degradation as network depth increases. To address these limitations, we propose stabilizing techniques for PCNs, including length regularization and sequential training with skip connection modules. This approach counteracts the exponential growth of latent states and makes the distribution of prediction errors more uniform across layers. Empirical evaluations demonstrate that our approach enhances stability and generalization, enabling the training of deeper networks more efficiently. This study deepens our understanding of complex dynamics in cortical networks, thereby advancing the practical application of predictive coding theory to its full potential.

## 1 INTRODUCTION

Predictive coding (PC) (Srinivasan et al., 1982; Mumford, 1992; Rao & Ballard, 1999) is a theoretical framework for cortical information processing, including learning, prediction, encoding, and memorization. Since PC theory accommodates key computations in the brain, it emerges as an appealing alternative for artificial intelligence (AI) systems aiming to achieve human-compatible learning and inference mechanisms (Salvatori et al., 2023a; Song et al., 2024). Deep predictive coding networks (PCNs) hold an untapped potential for engineering applications due to their inherent ability to perform local predictions and updates. This capability allows parallel computing during both learning and inference, making PCNs a viable alternative to conventional feed-forward neural networks that rely on backpropagation (Rumelhart et al., 1986). Additionally, predictive coding is adaptable to models with arbitrary topologies (Millidge et al., 2022b; Salvatori et al., 2022), making it a flexible framework for various learning tasks. That being said, critical issues limit our use of PCNs to their full potential: Training often becomes unstable, particularly when applied to deeper network architectures. The underlying causes of this instability are not fully understood, which limits the scalability and practical applicability of PCNs.

This study aims to address these challenges by analyzing the dynamics of latent state lengths during inference and proposing solutions to stabilize the training of deeper PCNs. The term length refers to the normalized squared norm, a more statistically traceable metric than the norm itself. Under the assumption of sufficiently high dimensionality, length approaches the variance of the elements of the respective variable. Since length is a norm-dependent metric, its changes are closely tied to variations in the norm. Specifically, changes in length during the dynamic processes of inference or learning can be used to detect variations in the norm while also providing insights into which layers exhibit a concentration of prediction error length. By leveraging dynamical mean-field theory (Sompolinsky et al., 1988; Poole et al., 2016; Schoenholz et al., 2017), we explore the recursive updates of latent state lengths in a random PCN ensemble and identify critical issues that hinder training.

1. We show that the latent state dynamics cause an exponential growth in the norms of latent states and prediction errors, directly impacting the gradients used for parameter updates. Such latent dynamics cause the exploding and vanishing gradient problems (EVGPs; (Bengio et al., 1994; Hochreiter, 1998; Pascanu et al., 2012; Arjovsky et al., 2016)) in PCNs.

2. We found that prediction errors tend to remain near the input and output layers after inference. Since prediction errors influence parameter updates, the occurrence of prediction errors predominantly near the input and output layer in deep networks prevents the other layers from being effectively trained. Consequently, this leads to a significant performance degradation as the network depth increases.

These issues are validated through a comparative analysis of theoretical predictions and empirical evidence.

To address the exponential growth of lengths, we explore regularization methods to suppress the pathological growth of the squared norms of the latent states. Additionally, to address the issue of prediction error occurring predominantly near the input and output layer, we introduce a sequential training framework with skip connection modules that enable the distributed propagation of prediction error. This approach, called stable prediction coding networks (SPCNs), effectively transfers the stalled prediction error to the other layers, stabilizing the learning process and improving performance. We validate the proposed methods through comprehensive analyses and experiments, demonstrating that PCNs with length regularization and skip connection modules significantly improve the stability and scalability of PCNs. Our approach guides the training of deeper predictive coding networks with improved accuracy and robustness.

**Overview**   This paper investigates key challenges and advancements in PCNs, focusing on theoretical and empirical insights. In Section 2, the background and motivation for this work are introduced by emphasizing a critical challenge in PCNs: achieving the inference condition $\Delta \mathbf{z} = 0$, as assumed in earlier studies, motivating a deeper exploration of this issue. Section 3 provides the theoretical framework for analyzing PCNs, presenting tools for length dynamics analysis. These tools form the foundation for understanding the underlying instabilities and inefficiencies in PCN inference processes. In Section 4, our observations regarding the dynamics of length in PCNs are then detailed, focusing on their inherent instability. The analysis reveals how instability in PCNs affects weight update lengths, contributing to the EVGPs. This finding underscores the difficulty of achieving the inference condition $\Delta \mathbf{z} = 0$. Additionally, the analysis shows that prediction errors are unevenly distributed across layers, primarily concentrating on the input and output layers. This uneven distribution allows intermediate layers to process information unrelated to the input or output, creating inefficiencies in learning and inference. In Section 5, we propose SPCNs to address these challenges. The proposed model introduces structural modifications that mitigate the identified issues, improving stability and performance. As shown in Section 6, experimental results validate the effectiveness of SPCNs in addressing the instability and error distribution problems outlined earlier. Finally, the paper discusses related works to situate this study within the broader context of predictive coding research and concludes by summarizing the contributions in the last sections.

## 2   FOUNDATIONS OF PREDICTIVE CODING NETWORKS

The Predictive coding (PC) theory posits that the brain continuously predicts the external environment and refines latent states by minimizing local prediction errors. A PCN, built based on the PC idea, connects local predictive coding modules in a chain structure. In this model, the forward step predicts the latent state of the subsequent layer, and the backward step minimizes local prediction errors, as shown in Figure 1. In contrast to the conventional von Neumann-style computational regime, the PC attempts to accommodate the human brain's cortical information processing with extremely parallel computation. They can also learn models with arbitrary graphical topologies (Millidge et al., 2022b; Salvatori et al., 2022).

**Neural Model**   The key characteristic of PCNs is that prediction and learning are performed locally. The prediction $\hat{\mathbf{z}}^{l+1}$ for the latent state of the $(l + 1)$-th layer $\mathbf{z}^{l+1}$ is made through the forward operation $f^l$ of the latent state of the $l$-th layer $\mathbf{z}^l \in \mathbb{R}^{N_l}$.

$$\hat{\mathbf{z}}^{l+1} = f^l(\mathbf{z}^l) = \phi(W^l \mathbf{z}^l + \mathbf{b}^l), \tag{1}$$

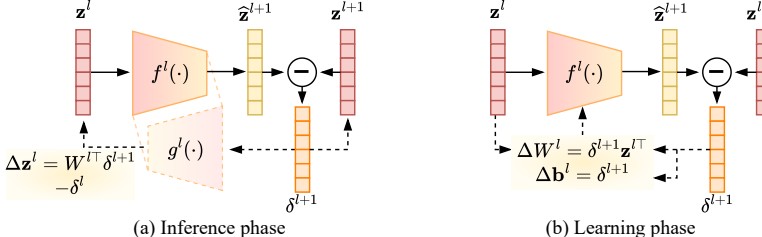

(a) Inference phase          (b) Learning phase

Figure 1: Inference and learning phases of a local predictive coding module. (a) During the inference phase, the prediction error $\delta^{l+1}$ is calculated to determine the changes in $\mathbf{z}^l$ and $\mathbf{z}^{l+1}$ according to Equation 2-3. This process is repeated until (ideally) the latent state converges or (practically) the final inference step $T$. The $\Delta \mathbf{z}^l$ equation assumes the linearity of $f^l$ and $g^l$. (b) During the learning phase, the weight and bias parameters are updated according to Equation 4-5. This learning phase begins once the inference phase is completed. The equations for $\Delta W^l$ and $\Delta \mathbf{b}^l$ are valid under the assumption of the linearity of $f^l$.

where $W^l \in \mathbb{R}^{N_{l+1} \times N_l}$ denotes the weight parameters, $\mathbf{b}^l \in \mathbb{R}^{N_l}$ represents the bias vector, and $\phi(\cdot)$ is a non-linear activation function, occasionally substituted with linearity in theoretical analyses (Whittington & Bogacz, 2017; Millidge et al., 2022a). The preactivation is denoted by $\mathbf{h}^l = W^l \mathbf{z}^l + \mathbf{b}^l$. The total and local PC losses are defined by $\mathcal{L}_{\text{PC}} = \sum_{l=2}^{L} \mathcal{L}_{\text{PC}}^l$ and $\mathcal{L}_{\text{PC}}^l = \frac{1}{2} \|\delta^l\|^2$, respectively, which allows the model to reduce the local prediction error for each layer $\delta^l = \mathbf{z}^l - \hat{\mathbf{z}}^l$ for $l \in [2, \ldots, L]$. In neuroscience, this concept often has been articulated as variational free energy minimization (Friston, 2005; Buckley et al., 2017). As an extension of Whittington & Bogacz (2017), we employ encoding-based PCNs with supervised learning settings; to learn input-output association $(\mathbf{x}, \mathbf{y})$, both ends of the latent layers are fixed to the corresponding data pair: $\mathbf{z}^1 = \mathbf{x}$ and $\mathbf{z}^L = \mathbf{y}$.

**Inference and Learning Process** To minimize prediction errors, the PCNs alternate the (latent state) inference and the learning phase; the former revises the latent states, while the latter adjusts the parameters (Figure 1). During inference, the model updates the latent states until the loss is sufficiently reduced. The learning phase with a single-step parameter update follows it. The latent state of each layer serves as a source of prediction for the subsequent layer, as well as a prediction target for the preceding layer. Accordingly, the latent states are updated as follows:

$$(1/\eta) \cdot \Delta \mathbf{z}^l = -\nabla_{\mathbf{z}^l} \sum_{k=2}^{L} \mathcal{L}_{\text{PC}}^k = -\nabla_{\mathbf{z}^l} \mathcal{L}_{\text{PC}}^l - \nabla_{\mathbf{z}^l} \mathcal{L}_{\text{PC}}^{l+1} = -\delta^l + g^l(\delta^{l+1}), \quad (2)$$

$$\mathbf{z}^{l,t+1} = \mathbf{z}^{l,t} + \Delta \mathbf{z}^{l,t}, \quad (3)$$

where $\eta$ represents the inference rate for latent update, $t$ denotes the inference step, and $g^l(\mathbf{a}) = {W^l}^\top D^l \mathbf{a}$ is the backward operation, where $D^l$ is $\text{diag}(\phi'(\mathbf{h}^l))$ if $\phi(\cdot)$ is nonlinear, or $I$ if linear. The latent state updates diminish the information of the $l$-th latent state. During the learning phase, the parameters are updated as follows.

$$(1/\zeta) \cdot \Delta W^l = -\nabla_{W^l} \sum_{k=2}^{L} \mathcal{L}_{\text{PC}}^k = -\nabla_{W^l} \mathcal{L}_{\text{PC}}^l = D^l \delta^{l+1} {\mathbf{z}^l}^\top, \quad (4)$$

$$W^{l,\tau+1} = W^{l,\tau} + \Delta W^{l,\tau}. \quad (5)$$

Here, $\zeta$ and $\tau$ refer to a learning rate and a learning step, respectively.

**Challenges in Achieving Equilibrium in Inference** The update rule of Equation 4 is analogous to error backpropagation of feed-forward networks (FFNs-BP). However, the distinctions between PCNs and FFNs-BP emerge in how the latent state $\mathbf{z}$ and prediction error $\delta$ are structured. The delta relation transforms into $(1/\eta) \cdot \Delta \mathbf{z}^l = 0 = -\delta^l + g^l(\delta^{l+1}) \implies \delta^l = g^l(\delta^{l+1})$, only if the model reaches the equilibrium state $\Delta \mathbf{z} = 0$, resembling $\delta$-propagation in BP. Based on this relation, the authors of Whittington & Bogacz (2017); Millidge et al. (2022a) argue that PCNs possess the same learning capabilities as BP. Nevertheless, our analysis indicates challenges in attaining the equilibrium through the inference phase (see Sections **??**), and even if reached, $\delta$ influences the forward mechanism of $\mathbf{z}$ unless $\delta$ equals 0 (i.e., $\mathbf{z}^l = f^{l-1}(\mathbf{z}^{l-1}) + \delta^l$). Thus, the update in PCNs diverges from FFNs-BP. As

$\delta \to 0$, the relation becomes similar to FFNs-BP only when learning concludes flawlessly. This makes it hard to understand the dynamics of inference and learning. Subsequent sections focus on analyzing the discrepancies in the lengths of the latent states, prediction errors, and parameters stemming from the update dynamics.

# 3   INVESTIGATING LENGTH DYNAMICS IN PREDICTIVE CODING NETWORKS

Unlike BP, PC involves an inference process, requiring a (tailored) analysis to understand its behavior. Using dynamical mean-field theory (Sompolinsky et al., 1988; Poole et al., 2016; Schoenholz et al., 2017), we examine the length dynamics of latent states during the inference process in PC ensemble models, where a length of a vector $\mathbf{x}$ is defined as the average of the squared entries of the vector (Poole et al., 2016), *i.e.*, $\langle x_i^2 \rangle = \frac{1}{N} \sum_{i=1}^{N} x_i^2$, where $\mathbf{x} = [x_1, \cdots, x_N]$ is a vector. In the inference phase, since $\Delta \mathbf{z}$ has not reached zero yet, it is challenging to describe the statistics of $\delta$ based on the relationship between $\delta^l$ and $\delta^{l+1}$ as in (Whittington & Bogacz, 2017; 2019; Millidge et al., 2022a). Therefore, using Equation 1-3, we expand the length dynamics in the recursive form of the latent state $\mathbf{z}$. These analyses play a fundamental role in understanding how information propagates and adapts across layers during inference.

**Assumptions**   In this simulation, elements of weights and biases are drawn i.i.d. as $w_{i,j}^l \sim \mathcal{N}(0, \frac{\sigma_w^2}{N_l})$ and $b_i^l \sim \mathcal{N}(0, \sigma_b^2)$. The elements of initial latent states at $t = 1$ are also drawn i.i.d.,: $z_i^l \sim \mathcal{N}(0, 1)$. To preclude any complication during the analyses, we assume the linearity of the prediction and homogeneous dimensionality of latent states across layers. For cases involving non-linear activations, empirical verification of the results is provided in Figure 2 and Appendix K.

**Update Rules of Latent States**   The forward and backward passes in the network can be expressed as $f^{l-1}(\mathbf{z}^{l-1}) = W^{l-1}\mathbf{z}^{l-1} + \mathbf{b}^{l-1}$ and $g^l(\mathbf{z}^{l+1}) = W^{l^\top}\mathbf{z}^{l+1}$, repectively. Given these transforms, we define the update rule for the latent state $\mathbf{z}^{l,t}$ as Equation 1-3. Expanding this rule, we obtain:

$$\mathbf{z}^{l,t+1} = \rho M^{l-1}\mathbf{z}^{l-1,t} + \kappa \mathbf{z}^{l,t} + \rho M^{l^\top}\mathbf{z}^{l+1,t} + \eta \mathbf{b}^{l-1} - \rho M^{l^\top}\mathbf{b}^l, \tag{6}$$

where $M = \frac{1}{\sigma_w}W$, $\rho = \eta\sigma_w$, and $\kappa = 1 - \eta(1 + \sigma_w^2)$. (See Appendix D.1 for the derivation.) This expression forms the basis for deriving the dynamics of latent states and interactions across layers.

**Capturing Layer-wise Interactions in PCNs**   We define several key matrices to model the interactions between latent states and biases over different layers. The *latent self-interaction* matrix $P_t$ captures the interaction between latent states at different layers: $P_{l+k,l}^t = \frac{1}{N}\mathbf{z}^{l+k,t^\top} M^{l+k-1:l}\mathbf{z}^{l,t}$, where $M^{l+k:l} = M^{l+k}M^{l+k-1}\cdots M^l$ is the product of normalized weight matrices across multiple layers. The diagonal elements of $P^t$ represent the length of the latent state at layer $l$, denoted as $p^{l,t}$, while off-diagonal elements describe cross-layer interactions. Additionally, we define the *bias-latent state interaction* matrix $B_t$ as: $B_{l,l-k}^t = \frac{1}{N}\mathbf{b}^{l-1^\top} M^{l:l-k}\mathbf{z}^{l-k,t}$. Finally, the *bias self-interaction* matrix $\Gamma$ captures interactions between biases at different layers and approaches to a constant matrix due to the fixed nature of bias terms and the distribution of Gaussian products with increasing dimensionality. Thus, we set $\Gamma = \sigma_b^2 I$.

**Analyzing the Dynamics of Interaction Matrices**   The interaction matrices evolve according to the latent state update rule. For the latent self-interaction matrix $P$, the update rule can be expressed as:

$$P_{l,l-k}^{t+1} = \mathbf{c}_z^\top P_{l-1:l+1,l-k-1:l-k+1}^t \mathbf{c}_z + \mathbf{c}_b^\top B_{l-k:l-k+1,l-1:l+1}^t \mathbf{c}_z$$
$$+ \mathbf{c}_b^\top B_{l:l+1,l-k-1:l-k+1}^t \mathbf{c}_z + \mathbf{c}_b^\top \Gamma_{l-1:l,l-k-1:l-k}\mathbf{c}_b,$$
$$P_{1,l}^{t+1^\top} = P_{l,1}^{t+1} = \mathbf{c}_z^\top P_{l-1:l+1,1}^t, \quad \text{and} \quad P_{L,l}^{t+1^\top} = P_{l,L}^{t+1} = \mathbf{c}_z^\top P_{l-1:l+1,L}^t, \tag{7}$$

for $1 < l, l - k < L$, where $\mathbf{c}_z = [\rho, \kappa, \rho]^\top$ and $\mathbf{c}_b = [\eta, -\rho]^\top$ are coefficient vectors, and $P_{a:b,c:d}$ is the submatrix of the $P$ matrix that spans from row $a$ to $b$, and from column $c$ to column $d$. This

equation accounts for interactions between latent states at different layers and time steps. Similarly, the bias-latent state interaction matrix $B$ is updated as:

$$B_{l,l-k}^{t+1} = B_{l,l-k-1:l-k+1}^t \mathbf{c}_z + \mathbf{c}_b^\top \Gamma_{l-k-1:l-k,l}. \tag{8}$$

The evolution of these matrices provides a comprehensive view of how latent states and biases interact during the inference process. (See Appendix D.4 for the derivation.)

**Analyzing the Length Dynamics** We now focus on the dynamics of lengths of latent states, prediction errors, and weight updates, denoted by $p^{l,t}$, $q^{l,t}$, and $r^{l,t}$, respectively. Each of the lengths can be traced through the derivations provided in Appendix D.5, utilizing the previously defined interaction matrices $P$, $B$, and $\Gamma$, or the values of other lengths referenced in the analysis. These lengths provide essential insights into the behavior of the variables as information propagates through the network. By the definition of the length, $p^{l,t} = \frac{1}{N}\mathbf{z}^{l,t^\top}\mathbf{z}^{l,t}$, which is is given by the diagonal elements of $P^t$ by definition of the interaction matrix, *i.e.*, $p^{l,t} = P_{l,l}^t$. The length of prediction errors at layer $l$ is $q^{l,t} = \frac{1}{N}\|\delta^{l,t}\|^2$ which can be expanded as a function of the latent state and bias interactions: $q^{l,t} = \mathbf{c}_q^\top P_{l-1:l,l-1:l}^t \mathbf{c}_q - 2B_{l,l-1:l}^t \mathbf{c}_q + \Gamma_{l-1,l-1}^t$, where the coefficient $\mathbf{c}_q = [-\sigma_w, 1]^\top$ (See Appendix D.5 for the derivation.). Finally, the length of the weight updates at each layer is $r^{l,t} = \frac{1}{N^2}\|\Delta W^l\|_F^2 = q^{l+1,t} \cdot p^{l,t}$, where $\Delta W_l$ is the weight update at layer $l$. The length of the bias update is equivalent to $q$ and is therefore omitted.

The length dynamics derived in this section are critical for understanding the temporal evolution of latent states and interactions in PCNs. By modeling latent state updates and their interactions with biases and neighboring layers, we have presented a comprehensive framework that captures the essential dynamics of inference in these networks. The analytical tools developed here provide a foundation for understanding the issues of length explosion and the skewed prediction error distribution in PCNs.

# 4 KEY CHALLENGES IN DEEP PCNS: INSIGHTS FROM LENGTH DYNAMICS

In this section, we identify key issues in the length dynamics observed in PCNs, focusing on the exponential growth of length dynamics (Subsection 4.1) and the concentration of prediction errors (Subsection 4.2).

## 4.1 EXPONENTIAL GROWTH IN LENGTH DYNAMICS

Figure 2 illustrates the length dynamics for $p^{l,t}$, $q^{l,t}$, and $r^{l,t}$ based on both analytical and empirical results.

**Length Depending on the Weight Variance:** The length of the latent state, $p^{l,t}$, is highly dependent on the weight variance, $\sigma_w$, while the impact of the bias variance, $\sigma_b$, is less significant (Figure 2a and i).

**Exponential Growth of Length Dynamics:** Figure 2e shows the mapping from $p^{l,t}$ to $p^{l,t+1}$, highlighting the transformation of latent states across inference steps. When the variance of the weight parameters is high ($\sigma_w = 5.4$), there is a noticeable tendency for $p^{l,t+1}$ to exceed $p^{l,t}$, as indicated by points above the line $p^{l,t+1} = p^{l,t}$. Conversely, when the weight variance is low ($\sigma_w = 0.185$), $p^{l,t+1}$ is less than $p^{l,t}$. Figure 2b illustrates the dynamics of $p^{l,t}$ over the inference step $t$. With $\sigma_w = 5.4$, $p^{l,t}$ grows exponentially, while for $\sigma_w = 0.185$, it decays exponentially. Importantly, the theoretical predictions (dotted lines) align closely with the experimental observations (solid lines). Figure 2f shows the layer-wise distribution of $p^{l,t}$ at the final inference step, $t = T$. The lengths of the latent states in the intermediate layers are significantly larger than those in the input and output layers, which are constant, especially for high values of $\sigma_w$. In a similar fashion to the latent state results, $q^{l,t}$ and $r^{l,t}$ exhibit exponential growth (Figure 2d and h), implying the potential for exploding and vanishing gradients of parameters by inference.

**Length Dynamics in Nonlinear PCNs:** In Figures 2c and g, we examine the latent state length dynamics in PCNs with nonlinear activation functions (tanh), confirming that the results are consistent with those observed for linear PCNs. One might expect that nonlinear activations would confine the latent state lengths, yet they directly impact the prediction values rather than confining the latent states.

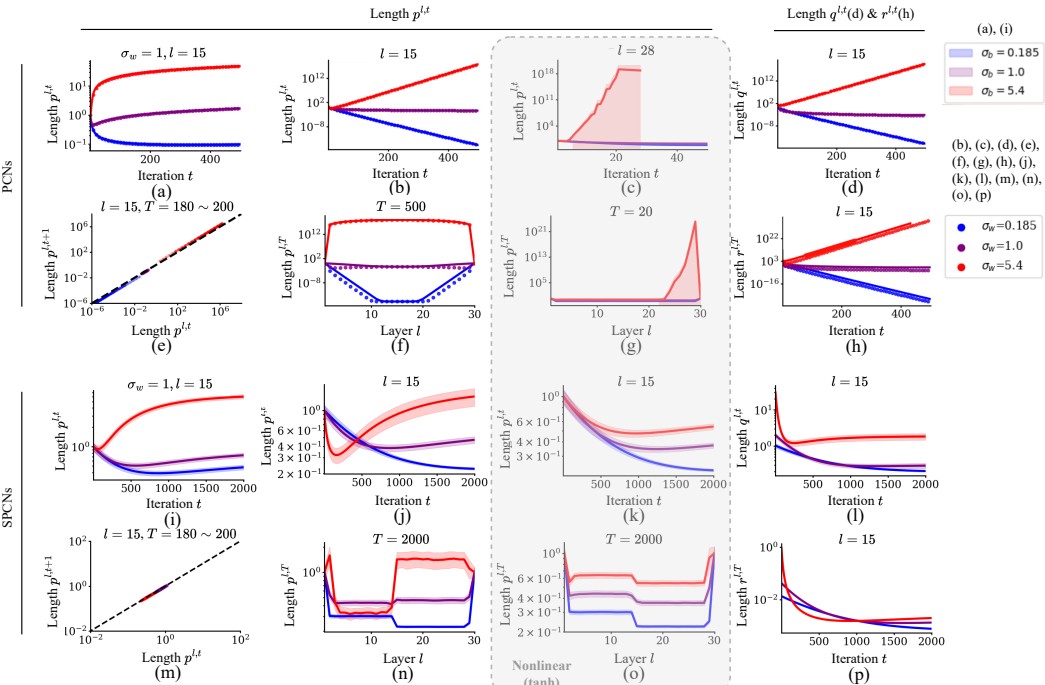

Figure 2: The dynamics of $p^{l,t}$, $q^{l,t}$, and $r^{l,t}$ and their layer-wise results for PCNs (a)-(h) and SPCNs (i)-(p). *Settings:* We set $L = 30$, with each layer having a latent dimension of $N = 100$. $\sigma_w^2$ and $\sigma_b^2$ represent the variance of the weights and biases, respectively. The inference rate is set to $\eta = 0.05$. (c), (g), (k), and (o) depict the results for nonlinear PCNs, while the remaining figures correspond to linear PCNs. Notably, the dotted lines represent theoretical predictions, while the solid lines correspond to empirical observations in the linear PCN results. In all subfigures, except for (a) and (i), the results are shown for the cases of $\sigma_w \in \{0.185, 1.0, 5.4\}$ with different colors, and $\sigma_b = 0$. (i)-(p) The experimental settings are identical to (a)-(h) but with length regularization applied. *Subfigures:* (a) Dynamics of $p^{l,t}$ across $\sigma_b \in \{0.185, 1.0, 5.4\}$. (b)-(d) & (h) Dynamics of $p^{l,t}$, $p^{l,t}$ of nonlinear PCNs, $q^{l,t}$, and $r^{l,t}$ for $T = 500$ inference steps. In (a), (b), (d), and (h), $l = 15$, while in (c), $l = 28$. (e) Scatter plot showing the points $(p^{l,t}, p^{l,t+1})$. The darker the dot's color, the later the inference step $t$. (f)-(g) Length of each layer at $t = 500$ and 20 for $p^{l,t}$ of linear and nonlinear PCNs, respectively. (i)-(p) The same subfigures as (a)-(h) but with length regularization applied.

The update process of the latent states involves changes in $\Delta \mathbf{z}$, which are influenced by the derivative of the nonlinearity. Still, this influence does not act as a confining mechanism. This growth pattern persists across various activation functions, as shown in Appendix K, indicating that nonlinearity alone does not fully mitigate the destabilizing effects. Additionally, this instability can emerge early in the inference phase, especially in deeper layers. These findings emphasize the importance of regularization strategies, such as those used in SPCNs, to prevent excessive growth and maintain stability in both linear and nonlinear PCNs, particularly when nonlinearity alone is insufficient to prevent the excessive growth of latent states and prediction errors.

In summary, the analysis reveals that both linear and nonlinear PCNs exhibit exponential growth in the length dynamics of latent states.

### 4.2 Concentrated Prediction Errors: How Prediction Errors Remain after Inference

**Information Propagation in Inference:** As described in Equation 6, the latent state at the inference step $t + 1$, denoted as $\mathbf{z}^{l,t+1}$, retains information from $\mathbf{z}^{l,t}$ at $\mathcal{O}(1)$, while prediction errors from adjacent layers are incorporated at $\mathcal{O}(\rho)$. By the inference step $t + 2$, this influence accumulates; $\mathbf{z}^{l,t+2}$ retains information from $\mathbf{z}^{l,t}$ at $\mathcal{O}(1)$, with influences from adjacent layers at $\mathcal{O}(\rho)$ and from layers two steps away at $\mathcal{O}(\rho^2)$. Without loss of generality, the influence from the $k$-th layer away is at $\mathcal{O}(\rho^k)$ after $k$-step updates.

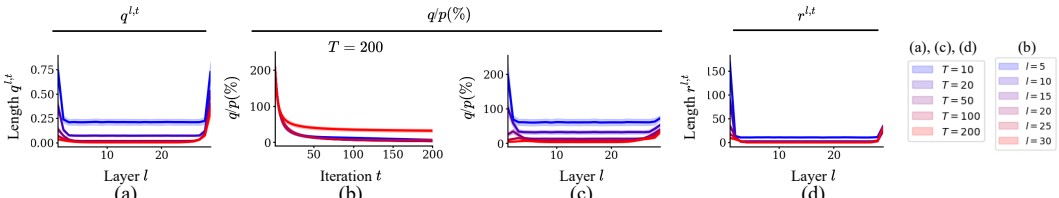

Figure 3: The length dynamics of the $q$ and $r$. *Setting:* We set $L = 30$, the terminal inference step is $T = 200$, and the latent dimension of each layer is set to 100. The inference rate is set to $\eta = 0.05$. $\sigma_w$ and $\sigma_b$ are set to 1 and 0.1, respectively. In (a), (c), (d), the cases for $t \in \{10, 20, 50, 100, 200\}$ are shown in different colors. In (b), the cases for layer $l \in \{5, 10, 15, 20, 25, 30\}$ are presented in different colors. In (c), the plots depict the $T = 200$ results. *Subfigures:* (a) The length of the prediction error, $q^{l,t}$ across layers at the inference step $t$. (b) The dynamics of $q/p$ over the inference step $t$ for each layer. (c) The ratio $q/p$ across layers at the inference step $t$. (d) The length of the amount of the weight parameter update, $r^{l,t}$, for each layer at the inference step $t$.

**Prediction Error Concentration:** As shown in Figure 3a, $q^{l,t}$ decreases in the intermediate layers with repeated iterations. In Figures 3b and c, to eliminate the influence of the latent state magnitude on the prediction error, since $\delta^{l,t} = \mathbf{z}^{l,t} - \hat{\mathbf{z}}^{l,t}$, we examine the ratio of $q^{l,t}$ to $p^{l,t}$. As inference proceeds, the $q/p$ ratio diminishes gradually, leaving significant normalized prediction errors concentrated near the input and output layers. As depicted in Figure 3c, after $T = 200$ inference steps, the $q/p$ ratio is significantly larger near the output layer. This trend also affects the updates of weight parameters after inference (Figure 3d).

**Impact of Concentrated Prediction Errors on Learning:** We observe that prediction errors tend to remain concentrated near the input and output layers during inference. According to Equation 4, updates to the weights and biases during subsequent learning phases become negligible if the prediction error approaches zero. While this issue has been previously addressed in Kinghorn et al. (2023), we demonstrate that the cause of this problem can be explained through length dynamics. In the following section, we propose a method to propagate the prediction error while maintaining the PC mechanism.

## 5 A Framework for Stable Learning in Deep Predictive Coding Networks

We propose SPCNs as a simple and practical solution to the challenges of PCNs. SPCNs incorporate length regularization to control the lengths of latent states and parameters, and sequential training with skip connection modules to enable progressive learning from layers close to the data pairs.

**Length Regularization** The exponential change in the length of the latent state affects the learning signal (see Section 4). We introduce length regularization to ensure stable update signals during training. This regularization directly applies to the latent states and learning parameters in the inference and learning phases. The regularization loss for a length $p$ is given by $\mathcal{R}_p := \sum_l (c_p - p^l)^2$, where $c_p$ indicates the target length of $p$. We similarly defined regularization loss for $r$, denoted by $\mathcal{R}_r$. Note that regularizing $q$ is equivalent to $\mathcal{L}_{\mathrm{PC}}$, and therefore it is omitted. By combining the regularization loss with the original predictive coding loss, the total loss becomes:

$$\mathcal{L} = \mathcal{L}_{\mathrm{PC}} + \lambda_p \mathcal{R}_p + \lambda_r \mathcal{R}_r, \tag{9}$$

where $\lambda$ is the corresponding length regularization loss coefficient. We set the target lengths as $c_p = 1$ and $c_r = 1$, and the coefficients as $\lambda_p = 1$ and $\lambda_r = 1$, respectively.

**Sequential Training with Skip Connection Modules** To address the issue introduced in Subsection 4.2, we propose a novel approach: sequential learning with a skip connection module, as illustrated in Figure 4. Details of this method are provided in Appendix E. We point out that prediction errors remain near the input or output layer after inference (depending on $T$). To mitigate this issue, we introduce skip-connection modules to directly propagate the prediction error from the output layer, thus preventing the latent states in intermediate layers from inferring and learning undesirable

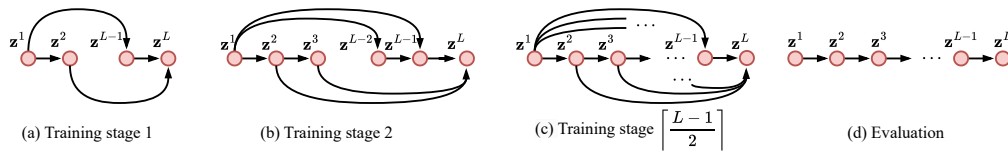

(a) Training stage 1      (b) Training stage 2      (c) Training stage $\left\lceil \frac{L-1}{2} \right\rceil$      (d) Evaluation

Figure 4: Sequential training with skip-connection modules. Each node represents a latent state, and each edge represents a local PC module (Figure 1). During the training phase, SPCNs include auxiliary skip connection edges in addition to the feed-forward connections. (a-c) Training is divided into $\lceil \frac{L-1}{2} \rceil$ stages, with (a), (b), and (c) representing stages 1, 2, and $\lceil \frac{L-1}{2} \rceil$, respectively. (d) Only the learned feed-forward connections are used for assessment in the evaluation phase.

Table 1: Lyapunov exponent of series of $p^{l,t}$ on MNIST dataset, comparing results with and without regularization (Rosenstein et al., 1993). The values are presented on a scale of $\times 10^{-2}$. N/A indicates that the Lyapunov exponent could not be computed due to the divergence of trajectories. Bold value (or text) indicates $\lambda_{\text{Lyapunov}} > 0$ (or N/A). $T = 500$ and $l = 15$.

| $\sigma_w$ | 0.1 | 0.2 | 0.5 | 1 | 2 | 5 | 10 |
|---|---|---|---|---|---|---|---|
| PCNs | -5.36 $\pm$ 4.29 | -3.34 $\pm$ 2.10 | -1.43 $\pm$ 2.58 | -2.08 $\pm$ 0.08 | -2.04 $\pm$ 0.12 | **2.38** $\pm$ 1.97 | **N/A** N/A |
| SPCNs | -2.10 $\pm$ 1.16 | -1.67 $\pm$ 1.43 | -0.64 $\pm$ 1.28 | -0.44 $\pm$ 0.78 | -1.66 $\pm$ 0.70 | -2.25 $\pm$ 0.10 | -2.29 $\pm$ 0.03 |

random predictions.

Conventional PCNs connect PC modules in a chain-like structure. Skip connection modules, on the other hand, serve as auxiliary PC modules connecting each latent state to the input or output directly, as shown in Figure 4. While these modules modify the chain-like structure of PCNs, each module maintains local inference and learning as a PC module.

The model gradually learns layer by layer toward the intermediate layers. The inference and learning process is divided into $\lceil \frac{L-1}{2} \rceil$ stages. The incremental process starts with the layer closest to the data pair and progressively includes increasingly distant layers. In the initial training stage, the latent states $\mathbf{z}^2$ and $\mathbf{z}^{L-1}$ closest to the data pair are inferred. The parameters associated with these states are learned in the subsequent learning phase. At this stage, the latent states from $\mathbf{z}^3$ to $\mathbf{z}^{L-2}$ do not participate in inference, and their connected predictive coding modules are not learned, isolating the information flow from input to output. It ensures that layers not involved in inference or learning do not integrate unnecessary information. However, this disconnection allows the latent states and parameters connected from the input to encode information independently of the output and vice versa.

During the inference phase at each stage, a relatively small $T$ is used to halt early, avoiding fitting the latent states to a prematurely trained network. Skip connection modules are applied only during training, while predictions during evaluation are made solely through the feed-forward connection modules.

## 6 EXPERIMENTS FOR SPCNS

The experiments in this section evaluate the stability and performance of SPCNs compared to (vanilla) PCNs. The study focuses on three key aspects: controlling the exponential growth of latent state lengths (Section 6.1), balancing prediction error distribution across network layers, and improving classification accuracy (Section 6.2).

### 6.1 CONTROLLED LENGTH DYNAMICS

**Length Dynamics Comparison** Controlled length dynamics are critical for ensuring stable and efficient performance in deep PCNs. We explore how SPCNs successfully manage length growth and maintain robust prediction performance. Figure 2i-p presents the dynamics of length for $p^{l,t}$, $q^{l,t}$, and $r^{l,t}$ within SPCNs. The length of the latent state, $p^{l,t}$, is influenced by the weight variance $\sigma_w$; however, this relationship is effectively regularized by SPCNs. In Figure 2m, the mapping between $p^{l,t}$ and $p^{l,t+1}$ demonstrates that SPCNs successfully constrain length dynamics near the

identity mapping line $p^{l,t+1} = p^{l,t}$, even for high variances ($\sigma_w = 5.4$). This contrasts with the behavior observed in standard PCNs. Additionally, Figure 2j illustrates that $p^{l,t}$ stabilizes, avoiding the exponential growth typically seen as $t$ increases. This stabilization is essential for maintaining manageable lengths throughout inference iterations. Figures 2n and o show that at the final iteration step $t = T$, SPCNs maintain relatively even lengths across layers, with intermediate layers no longer exhibiting excessive length expansion. Moreover, Figures 2k and o analyze SPCNs with nonlinear activation functions (tanh), demonstrating that the lengths of latent states remain constrained, in contrast to the divergence seen in standard nonlinear PCNs (Figure 2c and g). A similar trend can be observed for $q^{l,t}$ and $r^{l,t}$ under SPCNs, as shown in Figures 2l and p, where growth is significantly reduced compared to PCNs (Figure 2d and h).

**Lyapunov Exponent Comparison**    Furthermore, We compute the Lyapunov exponent to assess the stability of length dynamics. The Lyapunov exponent measures the rate of divergence or convergence of nearby trajectories in a dynamical system. It is defined as: $\lambda_{\text{Lyapunov}} = \lim_{t \to \infty} \frac{1}{t} \ln \frac{|\epsilon x(t)|}{|\epsilon x(0)|}$, where $\epsilon x(t)$ represents the separation between two initially close trajectories at time $t$. If $\lambda_{\text{Lyapunov}} > 0$, the system exhibits chaotic dynamics, with trajectories diverging exponentially. Using the Rosenstein method (Rosenstein et al., 1993), we estimate the largest Lyapunov exponent directly from empirical length data. The results, shown in Table 1, reveal that regularization reduces the Lyapunov exponent for $p^{l,t}$, $q^{l,t}$, and $r^{l,t}$ in SPCNs. For larger values of $\sigma_w$, SPCNs are still able to control the system with $\lambda_{\text{Lyapunov}} < 0$, whereas PCNs exhibit dynamics with $\lambda_{\text{Lyapunov}} > 0$. Length regularization ensures that the exponential growth of lengths across layers is suppressed, stabilizing inference.

**Length after Training**    Although our study primarily focuses on the length dynamics during inference, it is equally important to maintain stable length dynamics during training. Appendix J analyzes length dynamics after training, demonstrating that SPCNs effectively control the length dynamics during inference and learning.

6.2    SPCNs for Balanced Error Propagation

Table 2: Comparative results on MNIST and CIFAR-10. For the MNIST dataset, MLP-based PCNs/SPCNs are employed; For the CIFAR-10 dataset, CNN-based PCNs/SPCNs-s are employed. The table shows the classification performances (accuracy) of models based on different layer depths: $L = 3, 4, 6, 9$, and $13$. We evaluate models in terms of classification accuracy (%). Bold values represent the best performance for each configuration. Each entry provides the average accuracy (top) and one standard deviation (middle) from five runs. The table also reports the entropy (bottom) of the length of the prediction error, $q^{l,T}$. SPCN-s: SPCN without sequential training and skip connection modules from the input (See Appendix G).

| | | $L$ | 3 | 4 | 6 | 9 | 13 |
|---|---|---|---|---|---|---|---|
| MNIST | PCN (MLP) | acc. (%) | **90.61** $\pm\ 0.15$ | 88.81 $\pm\ 0.1$ | 83.31 $\pm\ 0.26$ | 81.17 $\pm\ 0.28$ | 49.97 $\pm\ 0.59$ |
| | | entropy | $5.09{\times}10^{-2}$ | $5.44{\times}10^{-2}$ | $5.33{\times}10^{-2}$ | $3.77{\times}10^{-3}$ | $5.91{\times}10^{-5}$ |
| | SPCN (MLP) | acc. (%) | 89.71 $\pm\ 0.18$ | **90.02** $\pm\ 0.06$ | **90.19** $\pm\ 0.21$ | **89.47** $\pm\ 0.18$ | **64.71** $\pm\ 1.46$ |
| | | entropy | $2.96{\times}10^{-2}$ | $2.31{\times}10^{-2}$ | $1.12{\times}10^{-2}$ | $1.02{\times}10^{-2}$ | $1.38{\times}10^{-2}$ |
| CIFAR-10 | PCN (CNN) | acc. (%) | **16.83** $\pm\ 0.64$ | 26.33 $\pm\ 0.44$ | 35.52 $\pm\ 0.42$ | **40.9** $\pm\ 1.06$ | 24.6 $\pm\ 6.06$ |
| | | entropy | $4.80{\times}10^{-6}$ | $7.68{\times}10^{-6}$ | $1.17{\times}10^{-73}$ | $7.46{\times}10^{-6}$ | $4.28{\times}10^{-3}$ |
| | SPCN-s (CNN) | acc. (%) | 16.36 $\pm\ 1.46$ | **27.69** $\pm\ 0.58$ | **45.1** $\pm\ 0.62$ | 37.65 $\pm\ 1.47$ | **29.12** $\pm\ 1.9$ |
| | | entropy | $3.80{\times}10^{-6}$ | $4.98{\times}10^{-5}$ | $2.17{\times}10^{-1}$ | $6.13{\times}10^{-2}$ | $5.29{\times}10^{-2}$ |

We conducted a series of experiments to verify the effectiveness of the proposed SPCN in addressing the discussed challenges.

**Balancing Errors and Classification Performance**    We trained both PCNs and SPCNs and evaluated their performance on the MNIST dataset. The detailed experimental setup is presented in Appendix F. We varied the layer depth $L$ from 3 to 13, measuring the test accuracy for each

configuration. Note that in this study, $L$ includes both the input and output layers. This experiment aimed to analyze how changes in network depth impact classification accuracy. As shown in Table 2, the classification performance of PCNs decreases with increasing layer depth $L$. In contrast, SPCNs demonstrated more stable performance across different layer depths, with smaller degradation as $L$ increased. Our experiments demonstrate that SPCNs effectively enhance performance in image classification tasks compared to traditional PCNs. To investigate whether prediction error concentration affects performance, we also measured entropy alongside accuracy by considering the lengths of the prediction errors as negative energy. The results in Table 2 suggest that in PCNs, entropy decreases as network depth $L$ increases, indicating that prediction errors concentrate on specific layers. In contrast, SPCNs maintain a relatively stable entropy scale, demonstrating a more even distribution of prediction errors across layers. In conclusion, introducing skip connection modules and sequential training in SPCNs addresses the prediction error concentration problem discussed in Section 4.2. SPCNs provide a more robust framework for predictive coding in deep networks by facilitating better error propagation and reducing bias toward the output layers.

**PC + CNNs on CIFAR-10** The performance of CNNs varies significantly depending on the latent state initialization schemes. While the feed-forward latent initialization scheme used in recent implementations of PCNs greatly improves performance, it does not align with the Gaussian assumptions made in this paper. In addition, it is not a fundamental component of the predictive coding mechanism. To demonstrate the performance of CNNs, we adopted the feed-forward latent initialization scheme. The resulting changes in the pattern of prediction error concentration and the corresponding modifications to SPCNs (SPCN-s) are discussed in Appendix G. Consequently, the performance advantage of utilizing CNNs alongside the feed-forward initialization scheme is somewhat attenuated.

Overall, SPCNs successfully address the challenges of exponential growth in length dynamics and prediction error concentration, offering a more robust and stable framework for deep PCNs.

## 7 CONCLUSION

This study delves into the underlying causes of unstable training in deep PCNs, focusing particularly on the length dynamics of the latent states during the inference process and the length of the parameter gradient affected by the dynamics. Our investigation highlights the exponential growth of the lengths of the latent states and prediction errors, culminating in the challenge of the exploding and vanishing gradient problem inherent to PCNs. Furthermore, we have identified that the intermediate layers of PCNs will likely be fitted to random predictions rather than information from data pairs, causing performance deterioration with increasing network depth. To address these issues, we introduce SPCNs, which implement strategies to regularize the lengths of latent states and adopt sequential training for deepening PCNs. Empirical evaluations of SPCNs demonstrate their efficacy in mitigating the issues and enhancing the trainability and generalizability of PCNs. Overall, this research significantly contributes to the advancement of PCNs by offering theoretical insights and empirical validation to improve their trainability and performance in practical applications.

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

## A    RELATED WORKS

A theoretical study (Millidge et al., 2022a) asserted that PCNs approximate learning rules such as backpropagation and target propagation, thereby highlighting the learning potential of PCNs. Previous studies (Whittington & Bogacz, 2017; 2019; Millidge et al., 2022a) have discussed the relationship with backpropagation under the assumption of equilibrium of latent states ($\Delta \mathbf{x} = 0$); however, we have directly probed the latent dynamics itself. Our analysis indicates that reaching the equilibrium can be challenging due to the exponential growth of transition in latent states. To our knowledge, this study is the first to investigate the inherent instability in the latent dynamics of PCNs. Our research is directly inspired by previous works (Poole et al., 2016; Schoenholz et al., 2017) that analyze stability issues in deep neural networks (DNNs) through the lens of length dynamics. These studies have focused on the stability of both forward and backward passes in DNNs, which is also linked to the EVGP (Bengio et al., 1994; Hochreiter, 1998; Pascanu et al., 2012; Arjovsky et al., 2016). We extend similar reasoning to demonstrate that this stability issue can also arise during the latent state inference process in PCNs. Previous research (Frieder & Lukasiewicz, 2022) explored the (non-)convergence behavior of PCNs, but our study identifies new challenges through an analysis of length dynamics. Specifically, we highlight the occurrence of EVGP from length analysis and the concentration of prediction errors in specific layers, forming the basis for our proposed sequential training with skip connection modules.

Meanwhile, PCNs have been applied to supervised and temporal sequence learning tasks (Whittington & Bogacz, 2017; Tang et al., 2023). However, although not explicitly stated, their published implementations of these studies are typically confined to a shallow network; generally, it is hard to train deep PCNs. Our study examines the stability and scalability issues inherent in deep PCNs through theoretical and empirical analysis. In addition, previous work (Kinghorn et al., 2023) has suggested that the instability of PCNs stems from discrepancies in updates across layers, which aligns with our observations regarding latent length dynamics. However, we approach this problem from a length dynamics perspective and propose SPCNs as a solution to enhance the entropy of prediction errors.

Predictive coding has also been adopted to design other models, such as  (Lotter et al., 2017; Choksi et al., 2021). PredNet (Lotter et al., 2017) predicts sequences by combining a PCN with Long Short-Term Memory (LSTM (Hochreiter & Schmidhuber, 1997)) units; however, unlike the original PC theory (Rao & Ballard, 1999), this approach relies on backpropagation rather than local computation and uses the encoded error as a prediction target. Predify (Choksi et al., 2021) represents a hybridization of FFNs-BP and PCNs, circumventing the chronic problems of PCNs by utilizing FFNs-BP and modern deep learning mechanisms such as batch normalization (Ioffe & Szegedy, 2015). In summary, many studies have adopted the predictive coding idea but often compromised the original theory. Our study pinpoints the issue underlying the unstable PCN training and improves its applicability to deeper PCNs.

## B    DISCUSSION

**Broader Impacts**    Predictive coding (PC) is a biologically plausible learning mechanism based on local computation. It acquired attention as a potential link between artificial network learning and brain learning (Whittington & Bogacz, 2017; 2019). However, we have demonstrated that PCNs have inherent limitations in stable learning. Our solution shows that regularization of neural activity is necessary for stable deep predictive coding in PCNs, which is supported by neuroscience research regarding the effects of normalization on the brain (Letellier et al., 2019. 6. 5.).
While our results advance our understanding of cortical computations, they could pose a risk of misuse, such as attacks or hacking of brain-like intelligence models. Finally, SPCNs still need to address fairness and privacy issues critical to various social concerns.

**Limitations**    We propose a stable, dynamic learning system and suggest an early stop of the inference phase to prevent random or premature overfitting of the predictive module. That said, its computational cost is still high compared to feed-forward neural networks, which may limit its practical applicability. The theory presented in our study is based on conventional PCNs with a feed-forward architecture, whereas the proposed model has a specific structure, as illustrated in Figure 4. Based on our theory, future works could suggest a more general theory accommodating

various structures, such as graph NNs (Wu et al., 2021) or transformers (Vaswani et al., 2023). We leave the theoretical understanding of complex topologies and their application to advanced cortical models for future research. Finally, our theoretical results mainly focus on the latent dynamics in random PCNs, leaving room for further research on the parameter dynamics.

## C  BIOLOGICAL PLAUSIBILITY OF SPCNs

### C.1  REGULARIZATION

Predictive Coding Networks (PCNs) have garnered attention for their alignment with cortical processes, given their reliance on local computations and the minimization of prediction errors—principles thought to underpin neural activity in the brain. In PCNs, each neuron updates its state based solely on local information, reflecting how neurons in the brain operate without a global control signal. This feature is particularly crucial for maintaining the biological plausibility of these models. However, traditional PCNs face challenges in terms of stability and trainability, raising questions about their biological plausibility when applied to deeper networks. The unstable learning dynamics observed in PCNs, characterized by the exponential growth of latent states and prediction errors, do not align with the stable learning observed in biological systems. This discrepancy suggests that while PCNs may capture some aspects of cortical function, they may not fully embody the robustness inherent in biological systems.

To address these issues, we propose Stable Predictive Coding Networks (SPCNs), which incorporate mechanisms to regularize the latent state dynamics, thereby improving stability and enhancing biological plausibility. SPCNs implement local computations similarly to traditional PCNs but introduce constraints that prevent the runaway dynamics observed in deeper networks. This modification aligns SPCNs more closely with the stable functioning of biological neural networks and enhances their scalability and applicability in practical settings. Despite these improvements, SPCNs retain some biological limitations. For instance, the introduction of regularization techniques, while stabilizing, may not have a direct analog in biological neural systems, where stability might be achieved through different, yet undiscovered, mechanisms. While SPCNs advance the biological plausibility of PCNs by addressing key stability issues, they represent an abstraction that balances biological fidelity with computational tractability. Further research is necessary to refine these models and explore how closely they can mimic the complex and dynamic processes observed in the brain.

### C.2  SQUENTIAL TRAINING

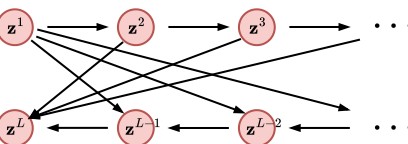

Figure 5: Sequential Training as Sequential Maturation

The SPCN framework draws inspiration from the nervous system's sequential maturation. In biological feedforward networks, supervised learning can be conceptualized as processing data pairs $(\mathbf{x}, y)$ as sensory information (Whittington & Bogacz, 2017). Our method, which prioritizes training layers closer to those handling external information such as data pairs, mirrors the sequential maturation observed in the brain as shown in Figure 5. For example, neural systems responsible for capturing low-level visual features mature earlier, akin to the hierarchical development of visual pathways in the nervous system.

Examples of sequential maturation in the nervous system include the following:

- **Early V1 Maturation**: During the early stages of visual cortex development, layer 4 of the primary visual cortex (V1), which receives input from the lateral geniculate nucleus (LGN), develops first. This leads to the formation of topographic maps and orientation selectivity (Espinosa & Stryker, 2012). Such foundational processing is completed before more complex visual processing develops in higher cortical areas.

- **Hierarchical Maturation of Visual Cortex**: Maturation signals are known to propagate hierarchically from primary sensory cortical regions to higher-order cortical processing areas (Chomiak & Hu, 2017). For instance, V1, which receives initial visual input, matures early to establish basic processing capabilities like edge detection. Subsequently, extrastriate areas (e.g., V2, V3, and V4) develop sequentially to support more complex visual tasks such as form recognition and color perception (Espinosa & Stryker, 2012; Mundinano et al., 2015; Siu & Murphy, 2018).

- **Development of Complex V1 Connections**: Research on the development of human V1 shows that feedforward connections from V1 to V2 mature earlier than feedback connections. Additionally, long-range horizontal connections within the superficial layers of V1, which integrate information across the visual field, continue to mature postnatally (Murphy & Monteiro, 2024) This suggests that circuits related to complex processing within V1 mature later than those responsible for basic input-output organization.

While the exact structural and maturational processes of the biological visual pathway are not directly emulated, PCNs are versatile and can be applied to various neural architectures (Salvatori et al., 2022). They exhibit varying degrees of biological plausibility (Mikulasch et al., 2023; Keller & Mrsic-Flogel, 2018; Salvatori et al., 2023b). The SPCN method draws inspiration from the brain's sequential maturation, facilitating stable long-term learning. By aligning with these principles, our method ensures consistency with predictive coding (PC) principles while maintaining stability and reinforcing its biological plausibility.

### C.3 SKIP CONNECTION MODULES

Our skip connection module mirrors structures observed in various neural circuits:

- **Visual Cortex Hierarchy**: The skip connection module reflects bypass connections in the visual cortex hierarchy (Blumberg & Kreiman, 2010; Hegdé & Felleman, 2007; Suzuki et al., 2023). For instance, the lower-level V1 directly connects to higher-order regions such as V4 or MT.

- **Thalamo-Cortical Loops**: The module aligns with circuits in the thalamocortical loop (Hegdé & Felleman, 2007; Suzuki et al., 2023), where regions such as the thalamus's LGN and pulvinar nuclei establish forward connections to multiple visual cortical areas.

- **Sensory-Motor Integration**: Skip connections link sensory information to motor outcomes, as seen in the pathways connecting the primary sensory cortex to higher cortical areas and subcortical motor centers (Suzuki et al., 2023).

These examples highlight that information in the cortex flows hierarchically and through lateral and recurrent pathways, enabling efficient and flexible error correction. Skip connection modules in PCNs emulate these pathways by directly propagating prediction errors across layers, improving both stability and error integration.

By incorporating skip connection modules, our SPCN model enhances computational stability while reflecting key cortical structure and function aspects. This design choice draws inspiration from hierarchical processing in the visual system and bypasses connections, thereby reinforcing the biological plausibility of our approach.

## D DERIVATION OF LENGTH DYNAMICS

This section investigates the length dynamics in predictive coding networks. We present the theoretical aspects of how latent variables evolve during the inference process, leveraging assumptions of linearity and Gaussian-distributed parameters. We focus on the statistical distribution of Gaussian samples, which serves as the foundation for understanding the Gaussian ensemble network's behavior under large-scale computations. We present a rigorous analysis of interaction matrices of latents and bias and their dynamics, offering insights into the lengths dynamics of latent states, prediction errors, and weight updates.

### D.1 ASSUMPTIONS AND LATENT STATES UPDATE RULE

Our primary goal is to track the changes in the length of the latent state during the inference step $t$. To perform this analysis, we adopt the following assumptions:

**Gaussian Assumption** We assume that the initial latent state at the inference step $t = 1$ is drawn i.i.d. as $z_i^{l,t} \sim \mathcal{N}(0,1)$. The learnable parameters, weight and bias, are drawn i.i.d. as $w_{i,j}^l \sim \mathcal{N}(0, \frac{\sigma_w^2}{N_l})$ and $b_i^l \sim \mathcal{N}(0, \sigma_b^2)$.

**Linearity Assumption** The correlation between variables may vary arbitrarily depending on the nonlinearity of the activation function $\phi$, making it difficult to expand interaction analytically. Therefore, we initially analyze the case where $\phi$ is a linear function. For cases involving non-linear activation functions, empirical verification is performed to confirm the results in Appendix K. In the context of the linearity assumption, the forward and backward transformations are defined as follows

$$f^{l-1}(\mathbf{z}^{l-1}) = W^{l-1}\mathbf{z}^{l-1} + \mathbf{b}^{l-1} \tag{10}$$

$$g^l(\mathbf{z}^{l+1}) = W^{l\top}\mathbf{z}^{l+1}. \tag{11}$$

**Dimensionality Assumption** We assume that all layers share the same dimensionality. If the dimensions differ, the latent spaces must be transformed using matrices like $M$, resulting in non-generalizable cross-layer interactions.

With these assumptions, we can expand the latent state update rule as follows:

$$\begin{aligned} \mathbf{z}^{l,t+1} &= \mathbf{z}^{l,t} + \Delta\mathbf{z}^{l,t} \\ &= \mathbf{z}^{l,t} + \eta\left(-\delta^{l,t} + W^{l\top}\delta^{l+1,t}\right) \\ &= \mathbf{z}^{l,t} + \eta\left(-\left(\mathbf{z}^{l,t} - \hat{\mathbf{z}}^{l,t}\right) + W^{l\top}\left(\mathbf{z}^{l+1,t} - \hat{\mathbf{z}}^{l+1,t}\right)\right) \end{aligned} \tag{12}$$

The update rule can be further simplified as:

$$\begin{aligned} \mathbf{z}^{l,t+1} &= (1-\eta)\mathbf{z}^{l,t} + \eta\cdot(W^{l-1}\mathbf{z}^{l-1,t} + \mathbf{b}^{l-1}) + \eta\cdot W^{l\top}\mathbf{z}^{l+1,t} - \eta\cdot W^{l\top}\left(W^l\mathbf{z}^{l,t} + \mathbf{b}^l\right) \\ &= \rho M^{l-1}\mathbf{z}^{l-1,t} + \kappa\mathbf{z}^{l,t} + \rho M^{l\top}\mathbf{z}^{l+1,t} + \eta\mathbf{b}^{l-1} - \rho M^{l\top}\mathbf{b}^l, \end{aligned} \tag{13}$$

where $M = \frac{1}{\sigma_w}W$, $\rho = \eta\sigma_w$, $\kappa = 1 - \eta(1 + \sigma_w^2)$, and $\xi = \eta\sigma_w^2$.

### D.2 THE DISTRIBUTION OF THE PRODUCT OF GAUSSIAN SAMPLES

Before delving into the dynamics of length, given that our analysis involves the product of different forms of Gaussian samples, it is essential to review the generalized results of this. Let $u_i \sim \mathcal{N}(0, \frac{\sigma_u^2}{N})$ and and $v_i \sim \mathcal{N}(0, \frac{\sigma_v^2}{N})$. The square of $u_i$ follows a chi-square distribution, while the product $u_i \cdot v_i$ follows a normal product distribution. Our interest lies in understanding the distribution of the following inner product values

$$\mathbf{u}^\top\mathbf{u} = \sum_{i=1}^N u_i^2 \quad\text{and}\quad \mathbf{u}^\top\mathbf{v} = \sum_{i=1}^N u_i \cdot v_i \tag{14}$$

as $N \to \infty$. Applying the Central Limit Theorem (CLT) to these values, we obtain the following:

$$\sqrt{N}\cdot\frac{\frac{\mathbf{u}^\top\mathbf{u}}{N}\mathbb{E}[u_i^2]}{\sqrt{\text{Var}(u_i^2)}} \to \mathcal{N}(0,1), \tag{15}$$

where $\mathbb{E}[u_i^2] = \text{Var}(u_i) = \frac{\sigma_u^2}{N}$, and $\text{Var}(u_i^2) = \mathbb{E}[u_i^4] - \mathbb{E}[u_i^2]^2 = 3\frac{\sigma_u^4}{N^2} - \frac{\sigma_u^4}{N^2} = 2\frac{\sigma_u^4}{N^2}$. As a result, $\mathbf{u}^\top\mathbf{u} \to \mathcal{N}(\sigma_u^2, \frac{2\sigma_u^4}{N})$, and equivalently,

$$\mathbf{u}^\top\mathbf{u} \to \sigma_u^2\cdot\mathcal{N}(1, \frac{2}{N}). \tag{16}$$

Similarly, applying the CLT to the cross-product yields:

$$\sqrt{N} \cdot \frac{\frac{\mathbf{u}^\top \mathbf{v}}{N} \mathbb{E}[u_i \cdot v_i]}{\sqrt{\text{Var}(u_i \cdot v_i)}} \to \mathcal{N}(0, 1), \tag{17}$$

where $\mathbb{E}[u_i \cdot v_i] = 0$, since $u_i$ and $v_i$ are independent, and $\text{Var}(u_i \cdot v_i) = \sigma_u^2 \cdot \sigma_v^2$. Hence, we obtain $\mathbf{u}^\top \mathbf{v} \to \mathcal{N}(0, \frac{\sigma_u^2 \cdot \sigma_v^2}{N})$. Equivalently, for large $N$,

$$\mathbf{u}^\top \mathbf{v} \sim \sigma_u \sigma_v \cdot \mathcal{N}(0, \frac{1}{N}), \tag{18}$$

and if $\sigma_u = \sigma_v$, this converges to $\sigma_u^2 \cdot \mathcal{N}(0, \frac{1}{N})$.

We can conduct a similar analysis for the distribution of vector lengths. Let $u_i \sim \mathcal{N}(0, \sigma_u^2)$ and $v_i \sim \mathcal{N}(0, \sigma_v^2)$. In these cases, we want to understand the asymptotic distribution of the following length terms:

$$\langle u_i^2 \rangle = \frac{1}{N} \sum_{i=1}^{N} u_i^2 \quad \text{and} \quad \langle u_i \cdot v_i \rangle = \frac{1}{N} \sum_{i=1}^{N} u_i \cdot v_i, \tag{19}$$

as $N \to \infty$. Note that the variance of the Gaussian distribution in the length calculation is not divided by $N$ in contrast to the inner product version. Instead, the length includes a division by $N$. By applying the CLT, similar to the inner product case, we have:

$$\langle u_i^2 \rangle \to \sigma_u^2 \cdot \mathcal{N}(1, \frac{2}{N}). \tag{20}$$

Using this result, we can apply it to the cases of interest.

**Lengths** In the case of vector-vector multiplication, consider vectors $\mathbf{z}^1$, $\mathbf{z}^L$, and $\mathbf{b}^l$, where $l \in \{1, \dots, L-1\}$. Each of these vectors is assumed to be sampled from a Gaussian distribution, i.e., each element is drawn from $\mathcal{N}(0, 1)$. The length defined by the relationship between these vectors, as $N \to \infty$, follows:

$$\langle u_i^2 \rangle \to \mathcal{N}(1, \frac{2}{N}), \tag{21}$$

while the cross-product between different vectors converges to:

$$\langle u_i \cdot v_i \rangle \to \mathcal{N}(0, \frac{1}{N}). \tag{22}$$

Consequently, the self-product (length) converges to 1, while the product with a different vector converges to 0 as $N \to \infty$. Finally, consider the length $l = \frac{1}{N} \mathbf{v}^\top A \mathbf{u}$, where each element of $A$, $A_{ij}$, is drawn from $\mathcal{N}(0, \frac{1}{N})$, and each element of $\mathbf{u}$ and $\mathbf{v}$ follows $\mathcal{N}(0, 1)$. The transformed vector $(A\mathbf{u})_i \sim \mathcal{N}(0, 1)$, Therefore, $\mathbf{v}^\top (A\mathbf{u}) \sim \mathcal{N}(0, 1)$, Thus, the length $l$ follows:

$$l \sim \mathcal{N}(0, \frac{1}{N^2}). \tag{23}$$

**Matrix-Matrix Multiplication** In the case of matrix-matrix multiplication, consider $C = A^\top A$, where each element of $A$, i.e., $A_{ij}$, is drawn from $\mathcal{N}(0, \frac{1}{N})$. The diagonal entries of $C$, $C_{ii}$, follow $\mathcal{N}(1, \frac{2}{N})$, The off-diagonal entries $C_{ij}$, where $i \neq j$, follow $\mathcal{N}(0, \frac{1}{N})$, Hence, $C$ approaches the identity matrix $I$ as $N \to \infty$. For the product of two matrices $D = AB$, where both $A_{ij}$ and $B_{ij}$ are sampled from $\mathcal{N}(0, \frac{1}{N})$, the resulting matrix $D_{ij}$ shares the same distribution as $A_{ij}$ and $B_{ij}$.

### D.3 INTERACTION MATRICES

For the analysis of length dynamics, we define several key variables as follows.

**Latent Self-Interaction** Let $P_{l+k,l}^t = \frac{1}{N}\mathbf{z}^{l+k,t\top}M^{l+k-1:l}\mathbf{z}^{l,t}$ for $1 \leq l, l-k \leq L$, where $M^{l+k:l} = M^{l+k}M^{l+k-1}\cdots M^l$ is the products of the series of matrices. By definition, we can observe that $P$ is systematic, meaning that $P_{l,l+k}^t = P_{l+k,l}^t$. The length of the latent state at layer $l$ and time step $t$, $p^{l,t}$ can be represented as the diagonal elements of $P^t$, $p^{l,t} = \left\langle \left(z_i^{l,t}\right)^2 \right\rangle = \frac{1}{N}\sum_{i=1}^{N}\left(z_i^{l,t}\right)^2 = \frac{1}{N}\mathbf{z}^{l,t\top}\mathbf{z}^{l,t} = P_{l,l}^t$. With this definition, the matrix $P^t$ contains the length information and interactions between latent states at different layers at the inference step $t$. Since the input and output are fixed during the inference phase as $\mathbf{z}^{1,t+1} = \mathbf{z}^{1,t}$ and $\mathbf{z}^{L,t+1} = \mathbf{z}^{L,t}$, the interaction terms with the indices 1 and $L$ are constants as $P_{1,1} = P_{L,L} = 1$ and $P_{1,L} = P_{L,1} = 0$. Similarly, at $t = 0$, $P^0 = I$.

**Bias-Latent State Interaction** Let bias-latent state interaction $B_{l,l-k}^t = \frac{1}{N}\mathbf{b}^{l-1\top}M^{l-1:l-k}\mathbf{z}^{l-k,t}$ be a bilinear term of interaction between the bias and latent states at layers $l$ and $l-k$ at inference step $t$ for $1 \leq l, l-k \leq L$. Likewise, let $B_{l-k,l}^t = \frac{1}{N}\mathbf{z}^{l,t\top}M^{l-1:l-k}\mathbf{b}^{l-k-1}$. Since the bias, the input ($\mathbf{z}^1$) and output ($\mathbf{z}^L$) are fixed during the inference phase, the interaction terms between these independent components, $B_{:,1}$ and $B_{:,L}$, are also fixed at 0. At $t = 0$, $B^0 = 0$.

**Bias Self-Interaction** The term $\Gamma_{l,l-k}$ represents $\frac{1}{N}\mathbf{b}^{l\top}M^{l:l-k+1}\mathbf{b}^{l-k} = 0$ for $1 \leq l, l-k \leq L$. Since the bias terms are sampled from $\mathcal{N}(0, \sigma_b^2)$ and fixed during the inference phase, $\Gamma = \sigma_b^2 I$ is a constant matrix by the properties introduced in Appendix D.2.

## D.4 DYNAMICS OF INTERACTION MATRIX

We derive the update rule for the $P$ using the definition of the interaction and the latent update rule in Equation 13. For an element of $P_{l,l-k}^t$, where $l-k > 1$ and $l < L$, the update equation can be described as follows:

$$
\begin{aligned}
P_{l,l-k}^{t+1} &= \frac{1}{N}\mathbf{z}^{l,t+1\top}M^{l-1:l-k}\mathbf{z}^{l-k,t+1} \\
&= \frac{1}{N}\left(\kappa\cdot\mathbf{z}^{l,t} + \rho\cdot M^{l-1}\mathbf{z}^{l-1,t} + \rho\cdot M^{l\top}\mathbf{z}^{l+1,t} + \eta\cdot\mathbf{b}^{l-1} - \rho\cdot M^{l\top}\mathbf{b}^l\right)^\top \\
&\quad \times M^{l-1:l-k} \\
&\quad \times \left(\kappa\cdot\mathbf{z}^{l-k,t} + \rho\cdot M^{l-k-1}\mathbf{z}^{l-k-1,t} + \rho\cdot M^{l-k\top}\mathbf{z}^{l-k+1,t} + \eta\cdot\mathbf{b}^{l-k-1} - \rho\cdot M^{l-k\top}\mathbf{b}^{l-k}\right)
\end{aligned}
$$

We want to expand this equation fully, showing all combinations of terms in the product. First, we identify the components of the vectors involved in the equation. The expression consists of a sum of transposed vectors, multiplied by a matrix $M^{l-1:l-k}$, and then multiplied by another sum of vectors. The components of the first sum of vectors are

$$
\mathbf{u}_1 = \rho\cdot M^{l-1}\mathbf{z}^{l-1,t}, \mathbf{u}_2 = \kappa\cdot\mathbf{z}^{l,t}, \mathbf{u}_3 = \rho\cdot M^{l\top}\mathbf{z}^{l+1,t}, \mathbf{u}_4 = \eta\cdot\mathbf{b}^{l-1}, \quad\text{and}\quad \mathbf{u}_5 = -\rho\cdot M^{l\top}\mathbf{b}^l.
$$

The components of the second sum of vectors are

$$
\mathbf{v}_1 = \rho\cdot M^{l-k-1}\mathbf{z}^{l-k-1,t}, \mathbf{v}_2 = \kappa\cdot\mathbf{z}^{l-k,t}, \mathbf{v}_3 = \rho\cdot M^{l-k\top}\mathbf{z}^{l-k+1,t}, \mathbf{v}_4 = \eta\cdot\mathbf{b}^{l-k-1},
$$
$$
\text{and}\quad \mathbf{v}_5 = -\rho\cdot M^{l-k\top}\mathbf{b}^{l-k}.
$$

We can rewrite the original equation using the components we defined:

$$
P_{l,l-k}^{t+1} = \frac{1}{N}\left(\mathbf{u}_1 + \mathbf{u}_2 + \mathbf{u}_3 + \mathbf{u}_4 + \mathbf{u}_5\right)^\top M^{l-1:l-k}\left(\mathbf{v}_1 + \mathbf{v}_2 + \mathbf{v}_3 + \mathbf{v}_4 + \mathbf{v}_5\right)
$$

We compute all possible products $\mathbf{u}_i^\top M^{l-1:l-k}\mathbf{v}_j$ for $i, j = 1$ to 5.

- Terms involving $\mathbf{u}_1$:

$$\mathbf{u}_1^\top M^{l-1:l-k}\mathbf{v}_1 = \rho^2 \left(\mathbf{z}^{l-1,t}\right)^\top M^{l-1^\top} M^{l-1:l-k}M^{l-k-1}\mathbf{z}^{l-k-1,t} = \rho^2 \cdot P_{l-1,l-k-1}^t$$

$$\mathbf{u}_1^\top M^{l-1:l-k}\mathbf{v}_2 = \rho\kappa \left(\mathbf{z}^{l-1,t}\right)^\top M^{l-1^\top} M^{l-1:l-k}\mathbf{z}^{l-k,t} = \kappa\rho \cdot P_{l-1,l-k}^t$$

$$\mathbf{u}_1^\top M^{l-1:l-k}\mathbf{v}_3 = \rho^2 \left(\mathbf{z}^{l-1,t}\right)^\top M^{l-1^\top} M^{l-1:l-k}M^{l-k^\top}\mathbf{z}^{l-k+1,t} = \rho^2 \cdot P_{l-1,l-k+1}^t$$

$$\mathbf{u}_1^\top M^{l-1:l-k}\mathbf{v}_4 = \rho\eta \left(\mathbf{z}^{l-1,t}\right)^\top M^{l-1^\top} M^{l-1:l-k}\mathbf{b}^{l-k-1} = \rho\eta \cdot B_{l-k,l-1}^t$$

$$\mathbf{u}_1^\top M^{l-1:l-k}\mathbf{v}_5 = -\rho^2 \left(\mathbf{z}^{l-1,t}\right)^\top M^{l-1^\top} M^{l-1:l-k}M^{l-k^\top}\mathbf{b}^{l-k} = -\rho^2 \cdot B_{l-k+1,l-1}^t$$

- Terms involving $\mathbf{u}_2$:

$$\mathbf{u}_2^\top M^{l-1:l-k}\mathbf{v}_1 = \kappa\rho \left(\mathbf{z}^{l,t}\right)^\top M^{l-1:l-k}M^{l-k-1}\mathbf{z}^{l-k-1,t} = \kappa\rho \cdot P_{l,l-k-1}^t$$

$$\mathbf{u}_2^\top M^{l-1:l-k}\mathbf{v}_2 = \kappa^2 \left(\mathbf{z}^{l,t}\right)^\top M^{l-1:l-k}\mathbf{z}^{l-k,t} = \kappa^2 \cdot P_{l,l-k}^t$$

$$\mathbf{u}_2^\top M^{l-1:l-k}\mathbf{v}_3 = \kappa\rho \left(\mathbf{z}^{l,t}\right)^\top M^{l-1:l-k}M^{l-k^\top}\mathbf{z}^{l-k+1,t} = \kappa\rho \cdot P_{l,l-k+1}^t$$

$$\mathbf{u}_2^\top M^{l-1:l-k}\mathbf{v}_4 = \kappa\eta \left(\mathbf{z}^{l,t}\right)^\top M^{l-1:l-k}\mathbf{b}^{l-k-1} = \kappa\eta \cdot B_{l-k,l}^t$$

$$\mathbf{u}_2^\top M^{l-1:l-k}\mathbf{v}_5 = -\kappa\rho \left(\mathbf{z}^{l,t}\right)^\top M^{l-1:l-k}M^{l-k^\top}\mathbf{b}^{l-k} = -\kappa\rho \cdot B_{l-k+1,l}^t$$

- Terms involving $\mathbf{u}_3$:

$$\mathbf{u}_3^\top M^{l-1:l-k}\mathbf{v}_1 = \rho^2 \left(\mathbf{z}^{l+1,t}\right)^\top M^l M^{l-1:l-k}M^{l-k-1}\mathbf{z}^{l-k-1,t} = \rho^2 \cdot P_{l+1,l-k-1}^t$$

$$\mathbf{u}_3^\top M^{l-1:l-k}\mathbf{v}_2 = \rho\kappa \left(\mathbf{z}^{l+1,t}\right)^\top M^l M^{l-1:l-k}\mathbf{z}^{l-k,t} = \kappa\rho \cdot P_{l+1,l-k}^t$$

$$\mathbf{u}_3^\top M^{l-1:l-k}\mathbf{v}_3 = \rho^2 \left(\mathbf{z}^{l+1,t}\right)^\top M^l M^{l-1:l-k}M^{l-k^\top}\mathbf{z}^{l-k+1,t} = \rho^2 \cdot P_{l+1,l-k+1}^t$$

$$\mathbf{u}_3^\top M^{l-1:l-k}\mathbf{v}_4 = \rho\eta \left(\mathbf{z}^{l+1,t}\right)^\top M^l M^{l-1:l-k}\mathbf{b}^{l-k-1} = \rho\eta \cdot B_{l-k,l+1}^t$$

$$\mathbf{u}_3^\top M^{l-1:l-k}\mathbf{v}_5 = -\rho^2 \left(\mathbf{z}^{l+1,t}\right)^\top M^l M^{l-1:l-k}M^{l-k^\top}\mathbf{b}^{l-k} = -\rho^2 \cdot B_{l-k+1,l+1}^t$$

- Terms involving $\mathbf{u}_4$:

$$\mathbf{u}_4^\top M^{l-1:l-k}\mathbf{v}_1 = \eta\rho \left(\mathbf{b}^{l-1}\right)^\top M^{l-1:l-k}M^{l-k-1}\mathbf{z}^{l-k-1,t} = \rho\eta \cdot B_{l,l-k-1}^t$$

$$\mathbf{u}_4^\top M^{l-1:l-k}\mathbf{v}_2 = \eta\kappa \left(\mathbf{b}^{l-1}\right)^\top M^{l-1:l-k}\mathbf{z}^{l-k,t} = \kappa\eta \cdot B_{l,l-k}^t$$

$$\mathbf{u}_4^\top M^{l-1:l-k}\mathbf{v}_3 = \eta\rho \left(\mathbf{b}^{l-1}\right)^\top M^{l-1:l-k}M^{l-k^\top}\mathbf{z}^{l-k+1,t} = \rho\eta \cdot B_{l,l-k+1}^t$$

$$\mathbf{u}_4^\top M^{l-1:l-k}\mathbf{v}_4 = \eta^2 \left(\mathbf{b}^{l-1}\right)^\top M^{l-1:l-k}\mathbf{b}^{l-k-1} = \eta^2 \cdot \gamma^{l-1,l-k-1}$$

$$\mathbf{u}_4^\top M^{l-1:l-k}\mathbf{v}_5 = -\eta\rho \left(\mathbf{b}^{l-1}\right)^\top M^{l-1:l-k}M^{l-k^\top}\mathbf{b}^{l-k} = -\rho\eta \cdot \gamma^{l-1,l-k}$$

- Terms involving $\mathbf{u}_5$:

$$\mathbf{u}_5^\top M^{l-1:l-k}\mathbf{v}_1 = -\rho^2 \left(\mathbf{b}^l\right)^\top M^l M^{l-1:l-k}M^{l-k-1}\mathbf{z}^{l-k-1,t} = -\rho^2 \cdot B_{l+1,l-k-1}^t$$

$$\mathbf{u}_5^\top M^{l-1:l-k}\mathbf{v}_2 = -\rho\kappa \left(\mathbf{b}^l\right)^\top M^l M^{l-1:l-k}\mathbf{z}^{l-k,t} = -\kappa\rho \cdot B_{l+1,l-k}^t$$

$$\mathbf{u}_5^\top M^{l-1:l-k}\mathbf{v}_3 = -\rho^2 \left(\mathbf{b}^l\right)^\top M^l M^{l-1:l-k}M^{l-k^\top}\mathbf{z}^{l-k+1,t} = -\rho^2 \cdot B_{l+1,l-k+1}^t$$

$$\mathbf{u}_5^\top M^{l-1:l-k}\mathbf{v}_4 = -\rho\eta \left(\mathbf{b}^l\right)^\top M^l M^{l-1:l-k}\mathbf{b}^{l-k-1} = -\rho\eta \cdot \gamma^{l,l-k-1}$$

$$\mathbf{u}_5^\top M^{l-1:l-k}\mathbf{v}_5 = \rho^2 \left(\mathbf{b}^l\right)^\top M^l M^{l-1:l-k}M^{l-k^\top}\mathbf{b}^{l-k} = \rho^2 \cdot \gamma^{l,l-k}$$

By systematically breaking down the original equation into its constituent components and computing all possible interactions between them, we have fully expanded the expression:

$$\begin{aligned}
P_{l,l-k}^{t+1} = {} & \rho^2 \cdot P_{l-1,l-k-1}^t + \kappa\rho \cdot P_{l-1,l-k}^t + \rho^2 \cdot P_{l-1,l-k+1}^t + \rho\eta \cdot B_{l-k,l-1}^t - \rho^2 \cdot B_{l-k+1,l-1}^t \\
& + \kappa\rho \cdot P_{l,l-k-1}^t + \kappa^2 \cdot P_{l,l-k}^t + \kappa\rho \cdot P_{l,l-k+1}^t + \kappa\eta \cdot B_{l-k,l}^t - \kappa\rho \cdot B_{l-k+1,l}^t \\
& + \rho^2 \cdot P_{l+1,l-k+1}^t + \kappa\rho \cdot P_{l+1,l-k}^t + \rho\eta \cdot B_{l-k,l+1}^t - \rho^2 \cdot B_{l-k+1,l+1}^t \\
& + \rho\eta \cdot B_{l,l-k-1}^t + \kappa\eta \cdot B_{l,l-k}^t + \rho\eta \cdot B_{l,l-k+1}^t + \eta^2 \cdot \gamma^{l-1,l-k-1} - \rho\eta \cdot \gamma^{l-1,l-k} \\
& - \rho^2 \cdot B_{l+1,l-k-1}^t - \kappa\rho \cdot B_{l+1,l-k}^t - \rho^2 \cdot B_{l+1,l-k+1}^t - \rho\eta \cdot \gamma^{l,l-k-1} + \rho^2 \cdot \gamma^{l,l-k} \quad (24)
\end{aligned}$$

On the other hand, when updating $P_{l,l-k}^t$, it is important to account for the cases where $l$ or $l-k$ are 1 or $L$, since the values of the latent states are fixed in such cases. For instance, the update equation for the interaction with the input layer, $P_{l,1}^{t+1}$, can be expressed as follows:

$$
\begin{aligned}
P_{l,1}^{t+1} &= \frac{1}{N}\mathbf{z}^{l,t+1\top} M^{l-1:1}\mathbf{z}^{1,t+1} \\
&= \frac{1}{N}\left(\kappa \cdot \mathbf{z}^{l,t\top} M^{l-1:1}\mathbf{z}^{1,t} + \rho \cdot \mathbf{z}^{l-1,t\top} M^{l-2:1}\mathbf{z}^{1,t} + \rho \cdot \mathbf{z}^{l+1,t\top} M^{l:1}\mathbf{z}^{1,t}\right) \\
&= \kappa \cdot P_{l,1}^t + \rho \cdot P_{l-1,1}^t + \rho \cdot P_{l+1,1}^t
\end{aligned}
\tag{25}
$$

Furthermore, by considering the symmetry of $p$, we have $P_{1,l}^t = P_{l,1}^t$. Similarly, the update equation for the interaction with the output layer, $P_{L,L-k}^{t+1}$, is as follows:

$$
\begin{aligned}
P_{L,L-k}^{t+1} &= \frac{1}{N}\mathbf{z}^{L,t\top} M^{L-1:L-k}\mathbf{z}^{L-k,t+1} \\
&= \frac{1}{N}\big(\kappa \cdot \mathbf{z}^{L,t\top} M^{L-1:L-k}\mathbf{z}^{L-k,t} + \rho \cdot \mathbf{z}^{L,t\top} M^{L-1:L-k-1}\mathbf{z}^{L-k-1,t} \\
&\quad + \rho \cdot \mathbf{z}^{L,t\top} M^{L-1:L-k+1}\mathbf{z}^{L-k+1,t}\big) \\
&= \kappa \cdot P_{L,L-k}^t + \rho \cdot P_{L,L-k-1}^t + \rho \cdot P_{L,L-k+1}^t
\end{aligned}
\tag{26}
$$

Moreover, $P_{L,L-k} = P_{L-k,L}$.

We now aim to express the above update rules, which involve many combination terms, in a more concise matrix and vector calculation form. Let us carefully examine the structure of the update equations for both the latent states and $p$. The update equation for $\mathbf{z}$ can be divided into two parts. The first part is the sum of the element-wise product of the latent states $[\mathbf{z}^{l-1}, \mathbf{z}^l, \mathbf{z}^{l+1}]^\top$ and the coefficients $\mathbf{c}_z = [\rho, \kappa, \rho]^\top$. The second part is the sum of the element-wise product of the bias terms $[\mathbf{b}^{l-1}, \mathbf{b}^l]$ and the coefficients $\mathbf{c}_b = [\eta, -\rho]^\top$. The update equation for $p$, which is derived from the update equation of $\mathbf{z}$, can be seen as the outer product of the latent updates of layer $l$ and another layer $l-k$. The coefficients are fixed, and the values of $l$ and $l-k$ correspond to the indices in the $P$ matrix. Utilizing this, we can rewrite the update rule from Equations 24 to 26 in matrix form as follows:

$$
\begin{aligned}
P_{l,l-k}^{t+1} &= \mathbf{c}_z^\top P_{l-1:l+1,l-k-1:l-k+1}^t \mathbf{c}_z + \mathbf{c}_b^\top B_{l-k:l-k+1,l-1:l+1}^t \mathbf{c}_z \\
&\quad + \mathbf{c}_b^\top B_{l:l+1,l-k-1:l-k+1}^t \mathbf{c}_z + \mathbf{c}_b^\top \Gamma_{l-1:l,l-k-1:l-k}^t \mathbf{c}_b, \\
P_{1,l}^{t+1\top} &= P_{l,1}^{t+1} = \mathbf{c}_z^\top P_{l-1:l+1,1}^t, \quad \text{and} \quad P_{L,l}^{t+1\top} = P_{l,L}^{t+1} = \mathbf{c}_z^\top P_{l-1:l+1,L}^t,
\end{aligned}
\tag{27}
$$

for $1 < l, l-k < L$.

The update rules for $B$ represent the evolution of the interaction between the latent states $\mathbf{z}$ and the bias terms $\mathbf{b}$.

$$
\begin{aligned}
B_{l+1,l-k}^{t+1} &= \frac{1}{N}\mathbf{b}^{l\top} M^{l:l-k}\mathbf{z}^{l-k,t+1} \\
&= \frac{1}{N}\mathbf{b}^{l\top} M^{l:l-k}(\kappa \cdot \mathbf{z}^{l-k,t} + \rho \cdot M^{l-k-1}\mathbf{z}^{l-k-1,t} + \rho \cdot M^{l-k\top}\mathbf{z}^{l-k+1,t} \\
&\quad + \eta \cdot \mathbf{b}^{l-k-1} - \rho \cdot M^{l-k\top}\mathbf{b}^{l-k}) \\
&= \frac{1}{N}\big(\kappa \cdot \mathbf{b}^{l\top} M^{l:l-k}\mathbf{z}^{l-k,t} + \rho \cdot \mathbf{b}^{l\top} M^{l:l-k}M^{l-k-1}\mathbf{z}^{l-k-1,t} \\
&\quad + \rho \cdot \mathbf{b}^{l\top} M^{l:l-k}M^{l-k\top}\mathbf{z}^{l-k+1,t} + \eta \cdot \mathbf{b}^{l\top} M^{l:l-k}\mathbf{b}^{l-k-1} - \rho \cdot \mathbf{b}^{l\top} M^{l:l-k}M^{l-k\top}\mathbf{b}^{l-k}\big) \\
&= \frac{1}{N}\big(\kappa \cdot \mathbf{b}^{l\top} M^{l:l-k}\mathbf{z}^{l-k,t} + \rho \cdot \mathbf{b}^{l\top} M^{l:l-k-1}\mathbf{z}^{l-k-1,t} \\
&\quad + \rho \cdot \mathbf{b}^{l\top} M^{l:l-k+1}\mathbf{z}^{l-k+1,t} + \eta \cdot \mathbf{b}^{l\top} M^{l:l-k}\mathbf{b}^{l-k-1} - \rho \cdot \mathbf{b}^{l\top} M^{l:l-k+1}\mathbf{b}^{l-k}\big) \\
&= \kappa \cdot B_{l+1,l-k}^t + \rho \cdot B_{l+1,l-k-1}^t + \rho \cdot B_{l+1,l-k+1}^t + \eta \cdot \Gamma^{l,l-k-1} - \rho \cdot \Gamma^{l,l-k}
\end{aligned}
\tag{28}
$$

We can simplify the update rule for $B$ as follows:

$$B_{l,l-k}^{t+1} = B_{l,l-k-1:l-k+1}^t \mathbf{c}_z + \mathbf{c}_b^\top \Gamma_{l-k-1:l-k,l}. \tag{29}$$

Note that since $\Gamma$ is a symmetric matrix, swapping the column and row indices in the update equation for $B$ does not alter the result.

### D.5 DYNAMICS OF LENGTHS

**Lengths of the Latent States** As mentioned earlier, the diagonal elements of $P_t$ represent the lengths of the latent states. That is, $p^{l,t} = P_{l,l}^t$.

**Lengths of the Prediction Errors** Let the length of the prediction error at layer $l$ be denoted by $q^{l,t}$. We can express it as follows:

$$q^{l,t} = \left\langle (\delta_i^{l,t})^2 \right\rangle$$

$$= \frac{1}{N} \|\delta^{l,t}\|^2$$

$$= \frac{1}{N} (\mathbf{z}^{l,t} - \hat{\mathbf{z}}^{l,t})^\top (\mathbf{z}^{l,t} - \hat{\mathbf{z}}^{l,t}), \tag{30}$$

where $\hat{\mathbf{z}}^{l,t} = W^{l-1}\mathbf{z}^{l-1,t} + \mathbf{b}^{l-1}$. By substituting the prediction term, $q^{l,t}$ can be further expanded as follows:

$$q^{l,t} = \frac{1}{N}(\mathbf{z}^{l,t} - \sigma_w \cdot M^{l-1}\mathbf{z}^{l-1,t} - \mathbf{b}^{l-1})^\top (\mathbf{z}^{l,t} - \sigma_w \cdot M^{l-1}\mathbf{z}^{l-1,t} - \mathbf{b}^{l-1})$$

$$= \frac{1}{N}(\mathbf{z}^{l,t\top} - \sigma_w \cdot \mathbf{z}^{l-1,t\top}M^{l-1\top} - \mathbf{b}^{l-1\top})(\mathbf{z}^{l,t} - \sigma_w \cdot M^{l-1}\mathbf{z}^{l-1,t} - \mathbf{b}^{l-1})$$

$$= \frac{1}{N}\big(\mathbf{z}^{l,t\top}(\mathbf{z}^{l,t} - \sigma_w \cdot M^{l-1}\mathbf{z}^{l-1,t} - \mathbf{b}^{l-1})$$

$$- \sigma_w \cdot \mathbf{z}^{l-1,t\top}M^{l-1\top}(\mathbf{z}^{l,t} - \sigma_w \cdot M^{l-1}\mathbf{z}^{l-1,t} - \mathbf{b}^{l-1})$$

$$- \mathbf{b}^{l-1\top}(\mathbf{z}^{l,t} - \sigma_w \cdot M^{l-1}\mathbf{z}^{l-1,t} - \mathbf{b}^{l-1}))$$

$$= \frac{1}{N}\big(\mathbf{z}^{l,t\top}\mathbf{z}^{l,t} - \sigma_w \cdot \mathbf{z}^{l,t\top}M^{l-1}\mathbf{z}^{l-1,t} - \mathbf{z}^{l,t\top}\mathbf{b}^{l-1}$$

$$- \sigma_w \cdot \mathbf{z}^{l-1,t\top}M^{l-1\top}\mathbf{z}^{l,t} + \sigma_w^2 \cdot \mathbf{z}^{l-1,t\top}M^{l-1\top}M^{l-1}\mathbf{z}^{l-1,t} + \sigma_w \cdot \mathbf{z}^{l-1,t\top}M^{l-1\top}\mathbf{b}^{l-1}$$

$$- \mathbf{b}^{l-1\top}\mathbf{z}^{l,t} + \sigma_w \cdot \mathbf{b}^{l-1\top}M^{l-1}\mathbf{z}^{l-1,t} + \mathbf{b}^{l-1\top}\mathbf{b}^{l-1}\big)$$

$$= P_{l,l}^t - \sigma_w \cdot P_{l,l-1}^t - B_{l,l}^t$$

$$- \sigma_w \cdot P_{l-1,l}^t + \sigma_w^2 \cdot P_{l-1,l-1}^t + \sigma_w \cdot B_{l,l-1}^t$$

$$- B_{l,l}^t + \sigma_w \cdot B_{l,l-1}^t + \gamma^{l-1,l-1} \tag{31}$$

The above equation can be simplified as:

$$q^{l,t} = \mathbf{c}_q^\top P_{l-1:l,l-1:l}^t \mathbf{c}_q - 2B_{l,l-1:l}^t \mathbf{c}_q + \Gamma_{l-1,l-1}^t, \tag{32}$$

where the coefficient $\mathbf{c}_q = [-\sigma_w, 1]^\top$.

**Lengths of Weight Updates** The length of the weight updates at layer $l$ is denoted by $r^{l,t}$, and is defined as:

$$r^{l,t} = \frac{1}{N^2}\|\Delta W^{l,t}\|_F^2, \tag{33}$$

where $\Delta W^{l,t} = \delta^{l+1,t}\mathbf{z}^{l,t\top}$, with $\delta^{l+1,t}$ representing the error signal at the next layer and $\mathbf{z}^{l,t}$ being the signal at the current layer.

Before proceeding further, we prove a simple lemma for the Frobenius norm:

**Lemma 1** *(Horn & Johnson, 2012)* $\|\mathbf{x}\mathbf{y}^\top\|_F^2 = \|\mathbf{x}\|^2 \cdot \|\mathbf{y}\|^2$, *where* $\mathbf{x}$ *and* $\mathbf{y}$ *are vectors.*

*Proof:*

$$\|\mathbf{x}\mathbf{y}^\top\|_F^2 = \sum_{i,j}(\mathbf{x}\mathbf{y}^\top)_{i,j}^2 = \sum_{i,j}(\mathbf{x}_i\mathbf{y}_j)^2 = \sum_i \mathbf{x}_i^2 \sum_j \mathbf{y}_j^2 = \|\mathbf{x}\|^2 \cdot \|\mathbf{y}\|^2. \tag{34}$$

Using Lemma 1 for the Frobenius norm, we can simplify $r^{l,t}$ as:

$$r^{l,t} = \|\delta^{l+1,t}\mathbf{z}^{l,t^\top}\|_F^2 = \|\delta^{l+1,t}\|^2 \cdot \|\mathbf{z}^{l,t}\|^2$$
$$= q^{l+1,t} \cdot p^{l,t} \tag{35}$$

Since $\frac{1}{N}\|\Delta\mathbf{b}^l\|^2 = \frac{1}{N}\|\delta^{l,t}\|^2$, The length of the bias update is equivalent to $q^{l,t}$ and is therefore omitted.

## E  DETAILS OF SEQUENTIAL TRAINING WITH SKIP CONNECTION MODULES

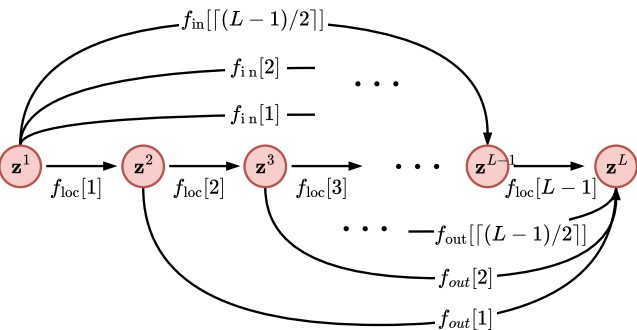

Figure 6: PC modules in Algorithm 1

In Algorithm 1, capital letters such as $X, Y,$ and $Z$ represent batch versions of the corresponding variables described in the main paper. The training process iterates $n_{\text{epochs}}$ epochs, which are divided into $n_{\text{stages}}$ stages. The lines 11-14 represent inference and 15-16 learning processes, repetively. Specifically, the algorithm incrementally expands the involvement of PC modules in inference and learning, starting near the input and output layers and gradually moving inward during each stage. For instance, the modules denoted as $f_{\text{loc}}$ compute the predictions and loss based on errors in the lines 24 and 26 in a chain structure. These computations span from the first to $i_{\text{stage}}$-th (line 24) and from $(L - i_{\text{stage}})$-th to $L$-th layers (line 26). These chain-based modules are complemented by additional skip connection modules denoted as $f_{\text{in}}$ and $f_{\text{out}}$ as in lines 29 and 30. Each of the modules denoted as $f_{\text{in}}$ is a module that takes $Z^1$ as input and predicts and compares with latent states from $(L - i_{\text{stage}})$-th to $(L - 1)$-th layers (line 29) . Each of the modules denoted as $f_{\text{out}}$ is a module that takes latent states from the second to $(i_{\text{stage}} + 1)$-th layers as input and predicts and compares with $Z^L$ (line 30). Figure 6 illustrates how the PC modules are connected and indicates their corresponding indices in the algorithm. This design effectively balances the computation of predictions and errors across layers, leveraging skip connection modules to maintain stability and accuracy during training.

## F  EXPERIMENTAL SETUP

This section details the experimental setup, code, and procedures for simulating the length dynamics and evaluating the proposed SPCNs and baselines. We provide comprehensive information to ensure reproducibility.

**Length Dynamics Analysis**  The simulations described in Section 4 analyzed the length dynamics of latent states and prediction errors during the inference process in a random PCNs ensemble. The

---

**Algorithm 1** Sequential Training with Skip Connection Modules

---

$n_{\text{epochs}}$: number of training epochs
$\mathcal{B}$: size of mini-batch
$T$: number of inference steps
$\mathcal{U}_z$: latent state update procedure
$\mathcal{U}_f$: PC module update procedure
$\mathcal{L}$: loss function based on prediction errors

1: **procedure** SEQUENTIALTRAINING_SKIPCONNECTIONMODULES
2:  Initialize PC modules $f_{\text{loc}}, f_{\text{in}}, f_{\text{out}}$
3:  $n_{\text{stages}} \leftarrow \lceil (L-1)/2 \rceil$
4:  $n_{\text{eos}} \leftarrow n_{\text{epochs}}/n_{\text{stages}}$
5:  **for** $i = 1 \cdots n_{\text{epochs}}$ **do**
6:   $i_{\text{stage}} \leftarrow (i + 1//n_{\text{eos}}) + 1$
7:   **for** $b = 1 \cdots \mathcal{B}$ **do**
8:    Sample a batch of data $X_b, Y_b$
9:    $Z^1 \leftarrow X_b, Z^L \leftarrow Y_b$
10:   $Z^{2:L-1} \sim \mathcal{N}(0, 1)$
11:    **for** $t = 1 \cdots T$ **do**
12:     loss $\leftarrow$ PREDICTIONLOSS($Z^{1:L}, i_{\text{stage}}$)
13:     $Z^{1:L} \leftarrow \mathcal{U}_z(\text{loss}, Z^{1:L})$
14:    **end for**
15:    loss $\leftarrow$ PREDICTIONLOSS($Z^{1:L}, i_{\text{stage}}$)
16:    $f_{\text{loc}}, f_{\text{in}}, f_{\text{out}} \leftarrow \mathcal{U}_f(\text{loss}, f_{\text{loc}}, f_{\text{in}}, f_{\text{out}})$
17:   **end for**
18:  **end for**
19:  **return** $f_{\text{loc}}$
20: **end procedure**

21: **procedure** PREDICTIONLOSS($Z^{1:L}, i_{\text{stage}}$)
22:  loss $\leftarrow 0$
23:  **for** $l = 1 \cdots i_{\text{stage}}$ **do**
24:   loss $+= \mathcal{L}(Z^{l+1}, f_{\text{loc}}[l](Z^l))$
25:   **if** $l < L - l$ **then**
26:    loss $+= \mathcal{L}(Z^{L-l+1}, f_{\text{loc}}[L-l](Z^{L-l}))$
27:   **end if**
28:   **if** $l < L//2$ **then**
29:    loss $+= \mathcal{L}(Z^{L-l}, f_{\text{in}}[l](Z^1))$
30:    loss $+= \mathcal{L}(Z^L, f_{\text{out}}[l](Z^{l+1}))$
31:   **end if**
32:  **end for**
33:  **return** loss
34: **end procedure**

---

dataset consisted of samples from a random unit Gaussian distribution ($(x_i, y_i) \sim \mathcal{N}(\mathbf{0}, I)$). The dataset contained 128 samples processed in a single batch. The number of inference steps ($T$) was mainly set to 2000 to track the iterative changes in length dynamics. The inference rate was fixed at 0.05. The model consisted of 30 layers to effectively show the exponential growth in PCN (Section 4). The latent dimension was set to 100.

**Image Classification** The experiments in Section 6 evaluate the classification accuracy of the proposed SPCN and baselines. Except for SPCN-specific parameters, all other parameters are identical to those used for the baseline PCNs. The dataset used is MNIST, with a batch size of 256. The inference steps were set to a short iteration number of $T = 10$ to prevent the overfitting of latent states in the intermediate layers. The inference rate decreases linearly from 0.2 to 0.02. We set the number of learning epochs to $L \times 5$. For example, when $L = 3$, the number of epochs is set to 15. The learning rate increases from 0 to 0.001 during the first 2% of total batch iterations as a warm-up

strategy. Subsequently, it follows stepwise learning rate decays, dropping by 0.1 at the 50% and 75% epochs of learning. The Adam (Kingma & Ba, 2017) optimizer was used to update the weights and biases with momentum parameters set to 0.9 and 0.999. The number of layers in the model varied from 3 to 13 to observe performance changes. For experiments where $L$ varied, $\sigma_w$ and $\sigma_b$ were fixed at 1.0 and 0.1, respectively. The activation function used was `tanh`.

**Common Setup** Each configuration was run five times using different seeds to ensure statistical robustness. Each run was performed on a single NVIDIA GeForce RTX 3090 GPU and completed within a few seconds. The implementation was done by using PyTorch (Paszke et al.). Our code is provided as supplementary material.

## G  FEED-FORWARD INITIALIZATION OF LATENT STATES

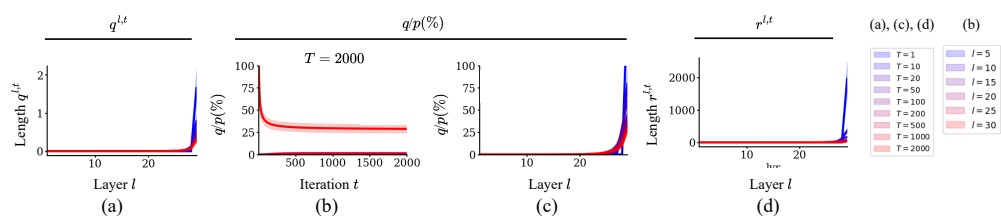

Figure 7: The length dynamics of the $q$ and $r$ with feed-forward latent initialization. *Setting & Subfigures:* same as in Figure 3.

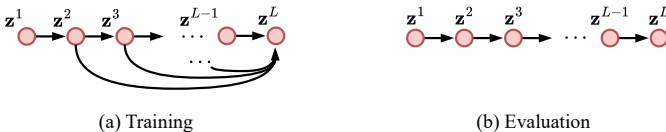

Figure 8: SPCN-s: a variation of an SPCN. (a) During the training phase, SPCN-s include feed-forward and auxiliary skip connections. (b) In the evaluation phase, only the learned feed-forward connections are used for assessment.

The performance of CNNs is highly dependent on the choice of latent state initialization schemes, with the feed-forward initialization scheme enhancing performance despite not aligning with the Gaussian assumptions of this paper. This section discusses the resulting changes in prediction error concentration patterns and the corresponding modifications to SPCNs following the adoption of this scheme.

**Prediction Error Concentration with Feed-Forward Initialization:** The performance of the network can vary significantly based on how the latent states are initialized, especially in the context of CNNs. Feed-forward initialization of latent states has been widely used in recent studies of PCNs (Whittington & Bogacz, 2017; Millidge et al., 2022b; Pinchetti et al., 2024). In this approach, $\mathbf{z}_{l,0} = f^{l-1}(\mathbf{z}_{l-1,0})$ for $2 \le l \le L-1$. Figure 7a shows that the length of the prediction error during inference is high near the output layer. At $t = 0$, since all latent states, except those at the output layer, are initialized by predictions, no prediction error occurs at the lower layers, with errors generated only at the output layer. According to Equation 2-3, latent states remain unchanged if no prediction error arises from adjacent layers. Thus, the latent state update begins with the output layer's prediction error. If a layer is $k$ layers away from the output layer, the prediction error propagates at the rate of $\mathcal{O}(\eta^k)$. Figure 7c shows that the length ratio near the output layer is significantly higher after $T = 2000$ inference steps. Similar to the case of Gaussian initialization, the length of the weight parameter update, $r^{l,t}$, follows a similar pattern (Figure 7d).

**Modification of SPCNs for Feed-Forward Initialization:** To address the issue of concentrated prediction errors near the output layer, we introduce modifications to the proposed SPCNs. As the initialization of latent states in a feed-forward manner necessitates the full connectivity of all feed-forward connection modules, the sequential training approach is rendered inapplicable. Furthermore,

due to the tendency of prediction errors to concentrate around the output layer, we restructure the skip connection modules to emphasize connections with the output layer, as depicted in Figure 8. Accordingly, in comparison to the original SPCNs, this revised configuration omits sequential training. It employs only a subset of skip connection modules directed towards the output layer, resulting in a less optimal variant. Consequently, the performance advantages typically associated with integrating the modified SPCNs and the feed-forward initialization scheme are moderately reduced.

## H  ABLATION STUDY

Table 3: Comparative results of an ablation study performed on MNIST over models from SPCN to PCN through its ablation. Each entry provides the average accuracy (top) and one standard deviation (middle) from five runs. This table follows the same format as Table 1. PCN+r: PCN with only regularization added to PCN. SPCN-sk: SPCN excluding skip connection modules. SPCN-sq: SPCN without sequential learning.

| L | | 3 | 4 | 6 | 9 | 13 |
|---|---|---|---|---|---|---|
| | acc. (%) | 90.69 | 89 | 83.29 | 81.46 | 50.49 |
| PCN | | $\pm 0.08$ | $\pm 0.2$ | $\pm 0.26$ | $\pm 0.15$ | $\pm 0.62$ |
| | entropy | $4.98 \times 10^{-2}$ | $5.20 \times 10^{-2}$ | $5.28 \times 10^{-2}$ | $3.74 \times 10^{-3}$ | $6.76 \times 10^{-5}$ |
| | acc. (%) | 90.71 | 88.81 | 83.37 | 81.63 | 50.98 |
| PCN+r | | $\pm 0.11$ | $\pm 0.29$ | $\pm 0.16$ | $\pm 0.11$ | $\pm 1.48$ |
| | entropy | $5.05 \times 10^{-2}$ | $5.25 \times 10^{-2}$ | $5.35 \times 10^{-2}$ | $3.72 \times 10^{-3}$ | $3.97 \times 10^{-5}$ |
| | acc. (%) | 89.43 | 85.91 | 81.35 | 49.69 | 22.64 |
| SPCN-sk | | $\pm 0.2$ | $\pm 0.21$ | $\pm 0.32$ | $\pm 1.24$ | $\pm 1.05$ |
| | entropy | $8.16 \times 10^{-2}$ | $2.93 \times 10^{-1}$ | $1.60 \times 10^{-1}$ | $4.55 \times 10^{-1}$ | $6.15 \times 10^{-1}$ |
| | acc. (%) | 89.57 | 90.12 | 90.44 | 89.39 | 64.95 |
| SPCN-s | | $\pm 0.11$ | $\pm 0.13$ | $\pm 0.13$ | $\pm 0.18$ | $\pm 2.30$ |
| | entropy | $3.20 \times 10^{-2}$ | $2.29 \times 10^{-2}$ | $1.21 \times 10^{-2}$ | $9.87 \times 10^{-3}$ | $1.40 \times 10^{-2}$ |
| | acc. (%) | 89.64 | 89.87 | 90.37 | 89.34 | 66 |
| SPCN | | $\pm 0.13$ | $\pm 0.18$ | $\pm 0.07$ | $\pm 0.25$ | $\pm 0.74$ |
| | entropy | $3.18 \times 10^{-2}$ | $2.24 \times 10^{-2}$ | $1.27 \times 10^{-2}$ | $9.88 \times 10^{-3}$ | $1.44 \times 10^{-2}$ |

We trained ablated models, including SPCN and PCN, and evaluated their performance on the MNIST dataset. The detailed experimental setup is provided in Appendix F. The experiment examined how variations in network depth and removing the proposed components affect classification accuracy and the entropy of prediction errors across the layer axis. As shown in Table 3, the classification performance of PCN declines as the network depth $L$ increases. While regularization provides marginal improvements, these gains are negligible compared to the improvements achieved by SPCN. In contrast, SPCN demonstrates more stable performance across various layer depths and significantly less degradation as $L$ increases. Notably, performance degradation in SPCN is minimal, even when sequential training is excluded. However, excluding skip connection modules leads to a substantial decline in performance. Sequential training without skip connection modules essentially trains on short-circuited chains, which perform poorly. Furthermore, entropy consistently correlates with performance, as SPCN and SPCN-sq demonstrate lower entropy alongside higher accuracy.

## I  ADDITIONAL RESULTS

Figure 9 shows the dynamics of the latent state lengths $p^{l,t}$, the prediction error lengths $q^{l,t}$, and the weight update lengths $r^{l,t}$ for PCNs across varying $\sigma_w$ and $\eta$ values. The results demonstrate how the network's stability depends heavily on the initialization of the weights and inference rate. In both the linear and nonlinear PCN cases, we observe that as $\sigma_w$ or $\eta$ increases, the system becomes more prone to instability, with the exponential growth of the latent state lengths becoming apparent. This is especially visible for higher values of $\sigma_w$ (e.g., 8.0), where the growth accelerates drastically. This behavior aligns with the theoretical predictions discussed in the paper, where weight variance $\sigma_w$ significantly influences the dynamics of the latent states. For smaller values of $\sigma_w$, such as 1.0, the growth is more contained, allowing the network to maintain more stable latent states across inference steps. However, larger values lead to a divergence in $p^{l,t}$, which necessitates additional regularization techniques, as suggested in our proposed SPCN framework.

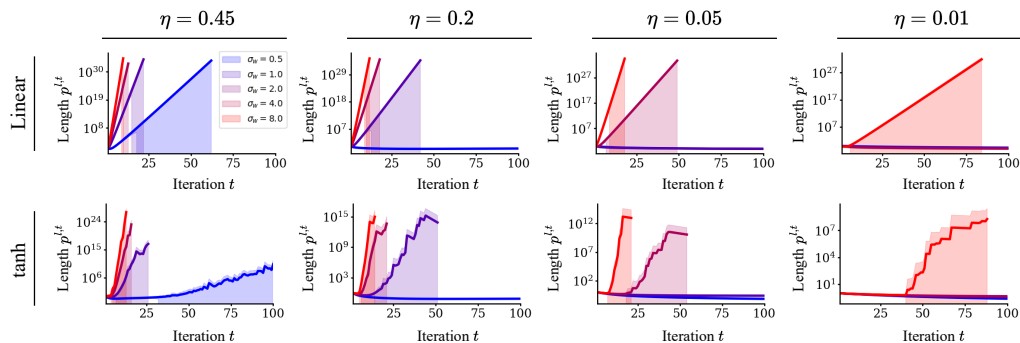

Figure 9: The dynamics of $p^{l,t}$, $q^{l,t}$, and $r^{l,t}$ for PCNs ($l = 26$ and $L = 30$). *Settings:* In all subfigures, the results are shown for the cases of $\sigma_w \in \{0.5, 1.0, 2.0, 4.0, 8.0\}$ with different colors, and $\sigma_b = 0.3$. Settings not mentioned or indicated are identical to those in Figure 2. *Subfigures:* (a)-(d) Dynamics of $p^{l,t}$ of linear PCNs over the 100 inference steps. (e)-(h) Dynamics of $p^{l,t}$ of nonlinear PCNs (tanh) over the 100 inference steps.

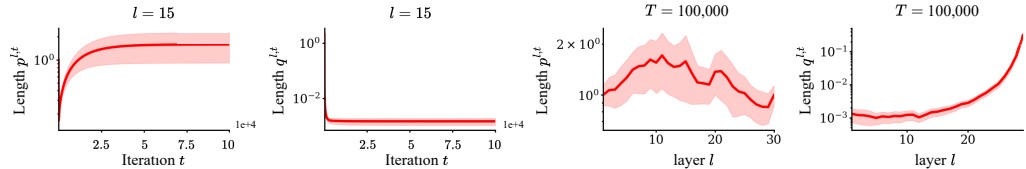

Figure 10: The dynamics of $p^{l,t}$ and $q^{l,t}$ for PCNs ($T = 100,000$). Settings not mentioned or indicated are identical to those in Figure 2.

Figure 10 explores the effect of extremely large inference steps ($T = 100,000$) on the dynamics of $p^{l,t}$ and $q^{l,t}$. Despite the large number of steps, the latent states and prediction errors stabilize after sufficient inference steps when $\sigma_w = 1.0$. However, we also observe that prediction errors tend to concentrate near the output layer, a phenomenon consistent with earlier findings that show concentrated prediction errors as a major challenge in deep PCNs. This stability over extended inference periods suggests that while PCNs can converge in theory, the issue of error concentration near the output layer persists. The results emphasize the need to balance prediction errors to prevent output-layer dominance, a feature crucial in deep networks for robust training.

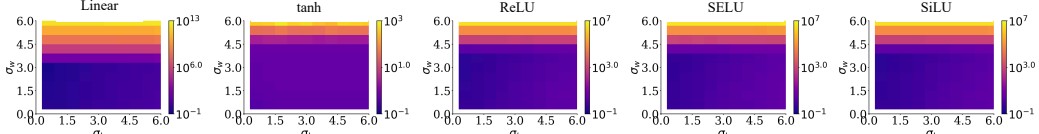

Figure 11: Heatmap plot of length $p^{l,t}$ for linear and nonlinear PCNs. ($\sigma_w \in \{0.6, 1.2, ..., 6.0\}$ and $\sigma_b \in \{0.6, 1.2, ..., 6.0\}$). The total number of inference $T = 10$ and the layer index $l = 15$. Settings not mentioned or indicated are identical to those in Figure 2.

Figure 11 provides a heatmap visualization showing the effects of $\sigma_w$ and $\sigma_b$ on the latent state lengths. For both linear and nonlinear PCNs, we observe that $\sigma_w$ has a much more significant impact on the length dynamics than $\sigma_b$. This supports the notion that the variance of the weights is the primary driver of instability, while the bias variance has a more subdued role. The heatmap also reveals that larger $\sigma_w$ values result in increasingly longer latent state lengths. These findings underline the necessity of controlling weight variance during initialization, as unchecked variance can lead to runaway growth in latent states.

Figure 12 presents a detailed examination of the layer-wise distribution of $p^{l,t}$, $q^{l,t}$, and $r^{l,t}$ in linear PCNs for different inference steps $t$. These subfigures aim to capture how the latent state lengths, prediction errors, and weight update magnitudes evolve across different layers and with varying $t$. In Figure 12a-c, for $\sigma_w = 5.4$, we observe a exponential growth pattern in the values of $p$, $q$, and $r$ across all layers, particularly as $T$ increases. This growth is expected, given that larger weight variances typically result in larger latent state dynamics, leading to a cascading effect on prediction

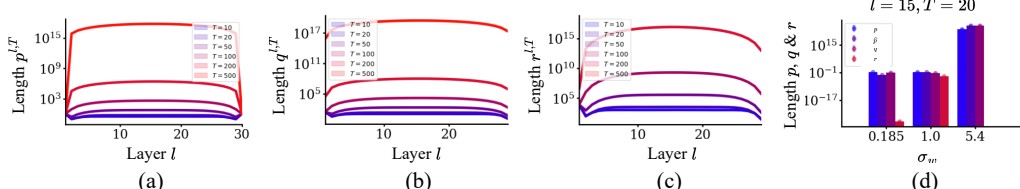

Figure 12: (a)-(c) The layer-wise distribution of $p^{l,t}$, $q^{l,t}$, and $r^{l,t}$ for linear PCNs ($\sigma_w = 5.4$ and $\sigma_b = 0.1$). The results are shown for the cases of $t \in \{10, 20, 50, 100, 200, 500\}$ with different colors. (d) Direct comparison of $p$, $\hat{p}$, $q$, and $r$ for $l = 15$ and $t = 20$. $\hat{p}$ represents the length of the prediction. Settings not mentioned or indicated are identical to those in Figure 2.

errors and weight updates. The increase in $p$, $q$, and $r$ with inference steps indicates that the internal representations become increasingly unstable as the inference phase progresses without proper regularization. Figure 12d highlights a direct comparison between $p$, $\hat{p}$ (the length of predictions), $q$, and $r$ for layer $l = 15$ at $T = 20$. Across all values of $\sigma_w$, we observe that $p$, $\hat{p}$, and $q$ remain within a similar range, though their values become more exaggerated for higher $\sigma_w$ values. Notably, $r$, which represents the weight update length, shows explosive growth when $\sigma_w = 5.4$, making it impractical to display fully. This behavior confirms that the higher values of $\sigma_w$ without regularization lead to unstable weight updates. Interestingly, for lower $\sigma_w$ values (e.g., $\sigma_w = 0.185$), $r$ remains small, indicating that proper initialization can contain these dynamics. However, $\sigma_w = 1$ shows a more moderate, controllable behavior in $r$. This figure emphasizes the need for length regularization and highlights the trade-off between network capacity (as influenced by $\sigma_w$) and the necessity of stability through regularization techniques.

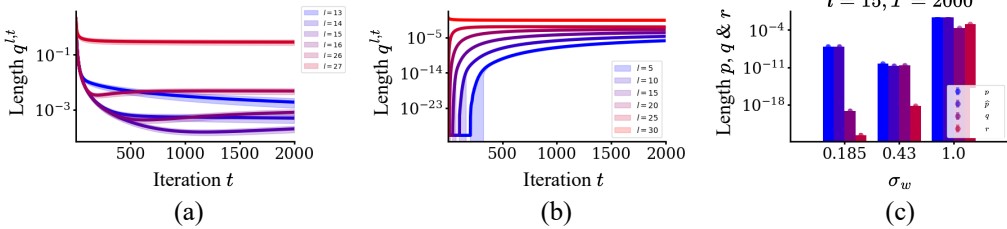

Figure 13: (a) & (b) The dynamics of $q^{l,t}$ for linear PCNs ($\sigma_w = 1$ and $\sigma_b = 0.1$). The results are shown for the cases of different layer index $l$ with different colors. (c) Direct comparison of $p$, $\hat{p}$, $q$, and $r$ for $l = 15$ and $T = 2000$. $\hat{p}$ represents the length of the prediction. Settings not mentioned or indicated are identical to those in Figure 3.

Figure 13 illustrates the dynamics of $q^{l,t}$ (prediction error lengths) in linear PCNs, with $\sigma_w = 1$ and $\sigma_b = 0.1$, across different layer indices and inference steps. In Figure 13a and b, we see that the prediction error length ($q^{l,t}$) increases significantly as we approach the output layer (indicated by red lines). This trend is consistent with the concentration of prediction errors in deeper layers, a challenge observed in deep PCNs that affects the learning capacity of intermediate layers. Conversely, the prediction error length in earlier layers (indicated by blue lines) starts small. It grows gradually with further inference steps, reinforcing the observation that early layers tend to stabilize more effectively than deeper layers. Figure 13c compares $p$, $\hat{p}$, $q$, and $r$ for layer $l = 15$ at $T = 2000$. The comparison shows how the dynamics of prediction lengths ($\hat{p}$), latent state lengths ($p$), and the magnitude of $r$ (weight update length) become highly dependent on $\sigma_w$. As noted earlier, the growth in $r$ with larger $\sigma_w$ values can lead to instability, stressing the need for controlled weight updates via regularization mechanisms.

## J  LENGTH ANALYSIS OVER TRAINING PROCESS

Figure 14 provides a comparison of the layer-wise latent state lengths ($p$) and prediction error lengths ($q$) between standard PCNs and SPCNs after the training process, for $\sigma_w = 5.4$. This figure serves as a direct contrast between the unregularized dynamics of PCNs and the stabilized dynamics in SPCNs. In the standard PCNs, $p$ and $q$ show exponential growth across layers, regardless of the value

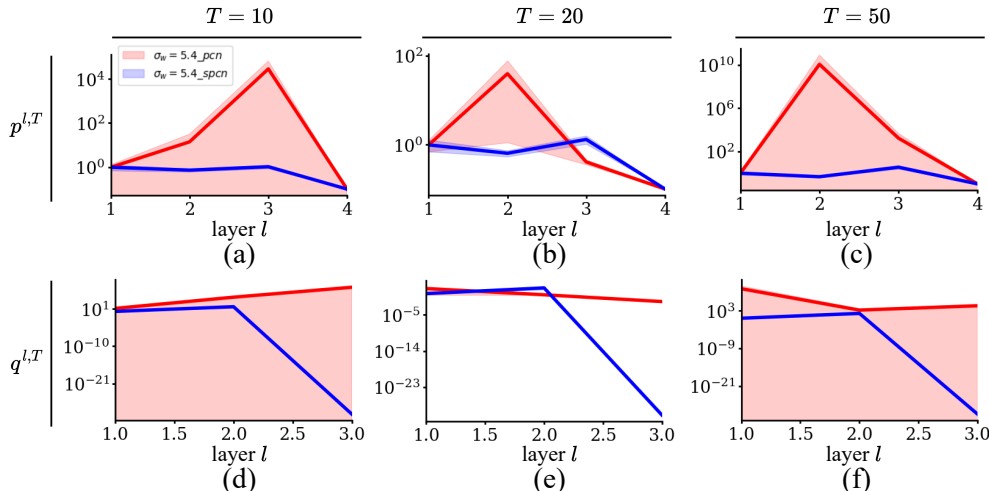

Figure 14: Layer-wise $p$ and $q$ comparison of PCN and SPCN ($\sigma_w = 5.4$). Settings are identical to those in Figure 3.

of $T$. This outcome is a direct consequence of the unregulated inference process, where latent states and prediction errors grow uncontrollably due to the lack of constraint on the interaction between layers. As a result, even early stopping during the inference phase fails to prevent length explosion, indicating that temporal constraints are insufficient for controlling latent dynamics. In contrast, the SPCNs exhibit a more contained growth of $p$ and $q$. Regularization techniques, as introduced in SPCNs, effectively prevent the exponential escalation of latent states and prediction errors across layers. The comparison demonstrates that without length regularization, PCNs struggle to maintain stable dynamics, while SPCNs ensure that the lengths remain within a manageable range, contributing to more stable and efficient training. Overall, this figure reinforces the necessity of regularization techniques, such as those employed in SPCNs, to control the growth of latent state lengths and prediction errors, thus ensuring the scalability and stability of the network during training.

## K  LENGTH DYNAMICS WITH NONLINEAR ACTIVATIONS

Figure 15 explores the dynamics of latent state lengths ($p^{l,t}$), prediction error lengths ($q^{l,t}$), and weight update lengths ($r^{l,t}$) in nonlinear PCNs across different activation functions. The analysis focuses on common nonlinearity types such as `tanh, ReLU, SELU,` and `SiLU`, each applied to layers with varying weight variances ($\sigma_w$). The results show that the dynamics for $p$, $q$, and $r$ are highly sensitive to $\sigma_w$, even in nonlinearity. The odd rows depict the temporal evolution of $p$, $q$, and $r$ at layer $l = 27$, while the even rows display the layer-wise distribution of these values at the $T = 20$ inference step. These subfigures illustrate two key phenomena that occur regardless of the applied nonlinear activation function:

1. In the odd rows, we observe that even with nonlinearity applied, $p$, $q$, and $r$ exhibit exponential growth near the output layer when $\sigma_w$ is large (e.g., $\sigma_w = 5.4$). This suggests that while nonlinear activations are typically expected to provide some degree of constraint on the predicting latent state dynamics by squashing the outputs (e.g., `tanh`), the latent state length growth persists for larger $\sigma_w$. This pattern holds across all activation functions examined, indicating that nonlinearity alone is insufficient to counteract the destabilizing effects of high weight variance.

2. The even rows reveal that these exponential growth patterns can emerge early in the inference phase, even at $T = 20$, particularly in deeper layers. The layer-wise distributions of $p$, $q$, and $r$ show that the effects of large $\sigma_w$ extend throughout the network, with prediction errors ($q$) and weight updates ($r$) becoming increasingly concentrated toward the output layer. This observation underscores a key challenge in training deep PCNs with nonlinearity. While early inference stages may seem stable, instability can rapidly accumulate in deeper layers due to the interplay between nonlinearity and large weight variances.

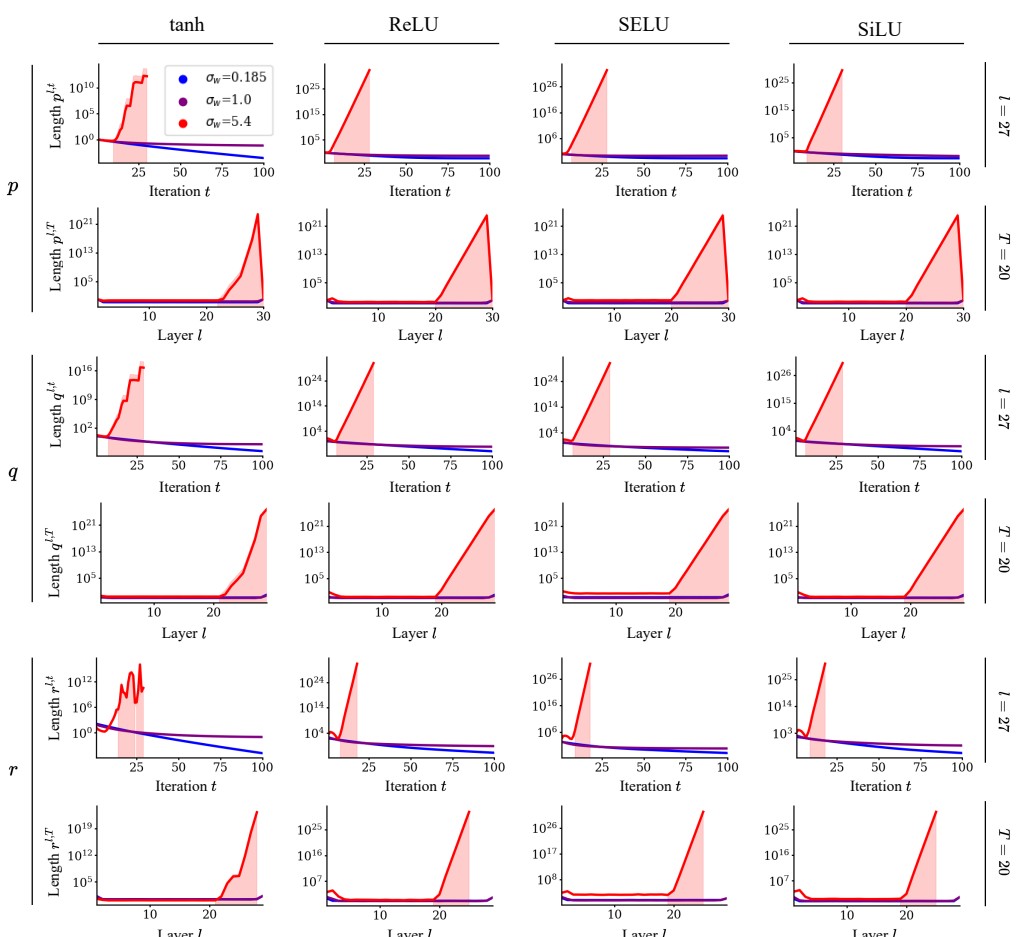

Figure 15: The dynamics of $p^{l,t}$, $q^{l,t}$, and $r^{l,t}$ and their layer-wise results for nonlinear PCNs. In all subfigures, the results are shown for the cases of $\sigma_w \in \{0.185, 1.0, 5.4\}$ with different colors, and $\sigma_b = 0.1$. Settings not mentioned or indicated are identical to those in Figure 2. Each column represents the applied nonlinear function. The odd rows are the dynamics of $p$, $q$, and $r$, respectively ($l = 27$). The even rows are the layer-wise distribution of $p$, $q$, and $r$, respectively ($T = 20$).

Importantly, this analysis highlights the need for regularization strategies, even in networks with nonlinear activations. The exponential growth seen here mirrors the behavior in linear PCNs, suggesting that length regularization and weight variance control are critical to preventing runaway dynamics in both linear and nonlinear architectures. Regularization techniques, such as those introduced in SPCNs, become essential for maintaining stability, particularly when nonlinearity alone is insufficient to prevent the excessive growth of latent states and prediction errors.

