# OpenReview forum: "Towards Stable Learning in Predictive Coding Networks"
_ICLR.cc/2025/Conference — Submitted to ICLR 2025_

### Official Review · Reviewer_vGUy · 2024-11-01

**Soundness:** 3
**Presentation:** 2
**Contribution:** 2
**Rating:** 3
**Confidence:** 5

**Summary:**

This work tackles an important problem in the literature of computational neuroscience and machine learning, that is, the scalability of computational models based on local message passing (in this case, predictive coding), to deep models and large scale tasks. In detail, they show that the cause of such problems lie in the norms of the of the latent variables being too large at the end of the model, and too small in the intermediate layers. The authors then address this problem via skip connections and a variation of the standard training framework, showing improvements in their empirical evaluation.

**Strengths:**

- The problem tackled is important: if we want to scale such models to complex performance, we need to understand the inherent problems of their dynamics. This may make interesting implications when it comes to implementations on neuromorphic hardware.

- The theoretical evaluation is quite interesting to me: I like how the authors have defined state interaction matrices, and the experimental study on how the proposed solution mitigates the problem. It is also notable how the theoretical predictions match the actual experiments.

- Skip connections are an interesting idea to address the issue (and lower the required number of iterations T needed to perform the inference phase).

**Weaknesses:**

1) The solution proposed, as well as the experimental evaluations, need improvement and argumentations:   The nice property of predictive coding, is the parallelisation of all of its operations, and the lack of external control that triggers them. As specified in [4]:   “Minimal external control. The neurons perform the computation autonomously with as little external control routing information in different ways at different times as possible.”

Is this requirement satisfied in your proposed solution? I suspect not, based on the description   “In the initial training stage, the latent states z2 and zL−1 closest to the data pair are inferred. The parameters associated with these states are learned in the subsequent learning phase. At this stage, the latent states from z3 to zL−2 do not participate in inference”  If not, I would suggest to adapt the proposed algorithm to perform all of the neuronal computations with no external control that specifies when specific layers have to be updated. Otherwise, if we could control when specific layers can be updated, we can turn PC into BP, and solve the stability problems [1,2]. To not mention that this is not only not acceptable from a biological plausibility perspective, but it also makes the model much harder to be implemented on neuromrphic chips.

2) Here is my biggest concern: the experimental evaluation for the image classification experiments. Why are the accuracies so low? On such small scale tasks, PCNs have been shown to be as good as backprop models: you can get >98% on MNIST using feedforward models, and >85% on CIFAR10 using convolutional ones (for the state of the art on the field, we refer to [5]). Is there an explanation on why they are so low?

3) Presentation could be improved: Add a pseudocode for the proposed algorithm, make the experimental sections clearer.

**Questions:**

The authors briefly mention similarities and differences between the weight update of PCNs and backprop. There are cases in which, by breaking the parallelism of the operations, and adding some external control that triggers the parameter updates at specific time steps, the updates of PC and BP are equivalent [1,2]. A discussion on this, and the other works you cite is [3].

Can the authors elaborate on the proposed SPCNs needing external control: what would be the advantage of such algorithm in practice?

Can the authors comment on the test accuracies obtained (for MNIST they are comparable with linear regression)?

The theoretical results are interesting, and so is the idea of addressing the problems of PCNs using the author’s findings. However, they could be used differently to improve the performance of deep PCNs. FOr example, I would focus on the skip connection idea (I believe there is value in it), and in the regularisation techniques proposed (how do they work if you apply them in isolation? This would be an interesting ablation study to add), without touching the autonomous updates. Furthermore, as the authors developed connections between the variance of the weights and the propagation of the error, how can PCNs be improved by changing the initialisation of the weights of intermediate layers?  The results seem to suggest that using initializations defined and optimized for BP is not the best way?


[1] Salvatori, Tommaso, et al. "Reverse differentiation via predictive coding." Proceedings of the AAAI Conference on Artificial Intelligence. Vol. 36. No. 7. 2022.

[2] Song, Yuhang, et al. "Can the brain do backpropagation?---exact implementation of backpropagation in predictive coding networks." Advances in neural information processing systems 33 (2020): 22566-22579.

[3] Rosenbaum, Robert. "On the relationship between predictive coding and backpropagation." Plos one 17.3 (2022): e0266102.

[4] Whittington, James CR, and Rafal Bogacz. "An approximation of the error backpropagation algorithm in a predictive coding network with local hebbian synaptic plasticity." Neural computation 29.5 (2017): 1229-1262.

[5] Pinchetti, Luca, et al. "Benchmarking Predictive Coding Networks--Made Simple." arXiv preprint arXiv:2407.01163 (2024).

---

### Official Review · Reviewer_HC7w · 2024-11-03

**Soundness:** 2
**Presentation:** 2
**Contribution:** 2
**Rating:** 5
**Confidence:** 3

**Summary:**

This paper proposes solutions to the known issue of instability in Predictive Coding Networks (PCNs) through length regularization and skip connection modules, resulting in the Stable PCN (SPCN). The authors use dynamical mean-field theory to analyze the length dynamics of latent states and attempt to stabilize training in PCNs. Their approach is evaluated on MNIST and CIFAR-10 benchmarks.

**Strengths:**

The paper tackles an important problem: the instability of training in PCNs, which has hindered their adoption despite their theoretical appeal. The paper provides a nice theoretical analysis of instability in PCN, and the proposed stabilizing techniques show empirical promise in some cases, notably MNIST.

**Weaknesses:**

The focus on length dynamics of latent states within the framework of dynamical mean-field theory is poorly justified. The authors do not explain why this framework is essential for addressing instability in PCNs or why "length dynamics" should be the primary focus. Although they reference previous works (Poole et al., 2016; Schoenholz et al., 2017), this is buried in the Related Work section, whereas the justification should be made upfront in the introduction. Without clear reasoning, this choice feels arbitrary and undermines the foundation of the paper, as the theoretical framework is crucial to supporting the proposed stabilizing techniques. The authors can move the relevant references from the Related Work section to the introduction and explicitly explain how this framework relates to addressing instability in PCNs. This would help readers understand the motivation and the underlying logic behind the chosen approach from the outset.

The authors introduce crucial mathematical quantities like $p^{l,t}$, $q^{l,t}$, and $r^{l,t}$ in lines 196-204 without providing sufficient intuitive explanations. The authors should highlight these equations and explain their meanings in a way that prepares the reader for the subsequent results.

The paper lacks a discussion on the interpretation of skip modules beyond their role in improving stability. Given the emphasis on the biological plausibility of PC, the authors must connect the use of skip modules to biological interpretations. Simply improving stability is not a strong enough argument for their inclusion. The authors should discuss the potential biological implications and interpretations of these modules, potentially drawing parallels with the prevalence of skip connections in the cortex and the non-hierarchical, recurrent organization of the brain ([Hegdé and Felleman, 2007](https://journals.sagepub.com/doi/10.1177/1073858407305201); [Suzuki et al., 2023](https://www.nature.com/articles/s41583-023-00756-z)). This would help tie their technical contribution back to the biological foundations of predictive coding theory.

The empirical results present a mixed picture, raising doubts about the reliability of the proposed SPCN method. The CIFAR10 benchmark shows inconsistencies, with PCN outperforming SPCN at certain network depths (even as deep as L = 9). The fluctuating accuracy as a function of depth, peaking at L = 6 layers, raises concerns about the consistency of the proposed method. In contrast to feedforward models with skip connections trained via backpropagation, which show consistent performance gains with added layers, the fluctuating performance in the CIFAR10 results raises questions about the reliability of the proposed SPCN approach. To address this issue, the authors should provide a more in-depth analysis of why their method shows fluctuating performance, particularly on CIFAR10. Discussing the potential reasons for the inconsistencies, and how these might be addressed in future work, could further enhance the paper.

The paper lacks a crucial baseline comparison: a PCN with simple L2 normalization on its weights or activations. Including this baseline would significantly strengthen the empirical evaluation by helping to determine whether the observed improvements are specific to the proposed SPCN mechanisms or simply due to the general effects of regularization.

**Overall Impression:**

While the paper addresses an important problem and presents interesting theoretical insights, the unclear presentation, mixed empirical results, and the lack of crucial baseline comparisons weaken the paper's impact. Therefore, I lean towards rejection. However, I am willing to reconsider my score if the authors address these concerns and provide a more compelling case for the significance of their work.

**Questions:**

See weaknesses.

---

### Official Review · Reviewer_AXyX · 2024-11-03

**Soundness:** 3
**Presentation:** 2
**Contribution:** 2
**Rating:** 6
**Confidence:** 3

**Summary:**

The paper analyzes the reasons behind training instability in predictive-coding networks (PCN), identifying in particular exponential growth in the latent states and concentration of prediction errors around input and output as problems. The authors propose two stabilizing techniques: length regularization, that tries to keep the norms of the latent states and of the changes in connection weights close to a set point; and sequential training with skip modules, which runs inference in multiple stages, at each stage including only some of the first and last layers in the network, with skip connections between them. Experiments show that this approach (that they dub stable PCN, SPCN) can help maintain stability and performance in deeper networks.

**Strengths:**

**Originality**
* the paper introduces an interesting new way of training PCNs, by first choosing the first two nodes and last two nodes and connecting them using a skip connection; then adding in another node and another skip connection; etc.

**Quality**
* the paper includes extensive analytical derivations in addition to simulations.

**Clarity**
* The introduction to PCN is pretty clear, though the rest of the paper could be improved in clarity.

**Significance**
* understanding how brains learn is important, and the techniques currently employed for training artificial neural networks are not particularly plausible in the biological context. Predictive coding is one of the main contenders for a biologically plausible learning rule, but has the downside of not being as powerful as backpropagation at the moment. As such, any work attempting to improve on this is highly relevant.

**Weaknesses:**

* The paper as written is a bit too dense and does not sufficiently explain the bigger picture.
* I'm having a hard time following exactly how the sequential training works:
  * Do the skip connections just act like usual predictive-coding connections (except with a fixed weight)? Or how do they factor into the inference and dynamics?
* Has any optimization of hyperparameters been attempted? Otherwise it seems hard to compare different models.
* Fig. 3: I don't see the point of using 9 values for $T$ and 6 values for $l$; at most 4-5 $T$ values are visible in the panels, and perhaps 2-3 $l$ values. I suggest using fewer values to make the legend easier to parse.

**Questions:**

* From Table 2, I can see that the SPCN performance is arguably better than plain PCN; however, it's far from stellar, and it still degrades for deeper nets, so that brings up the question of how significant is this?
  * in particular, for the CIFAR10 example you are using CNNs, but in a biological context having repeated weights as in a convolutional net is difficult to implement. If we're not bio plausible anyway, why use predictive coding?
* I can't really parse these sentences around lines 121-122: "The latent state updates diminish the information of the $l$th latent state. It is redeemed by the latent information of the neighboring layers transformed to the $l$th layer’s latent space." If this is important for the rest of the paper, it needs to be expanded; otherwise, it's probably best to remove it.

---

> ### Comment · Reviewer_AXyX · 2024-11-26
>
> Thanks for your answers! I increased my score to 6 (and the confidence to 3) as a result of the clarifications and updates to the manuscript. I do however think that hyperparameter optimization is essential when comparing different models. Otherwise it's impossible to tell whether any advantages of the proposed method are due to the method itself or simply because of a more fortuitous choice of hyperparameters.

---

### Official Review · Reviewer_vHcq · 2024-11-04

**Soundness:** 2
**Presentation:** 2
**Contribution:** 1
**Rating:** 1
**Confidence:** 5

**Summary:**

In this, the author proposes SPCN that uses skip connection and early stopping to stabilize the learning in the predictive coding network, which helps the model learn when network depth increases. Experiments are performed on toy datasets across layers such as mnist and cifar-10 to show model behavior.

**Strengths:**

Scaling PC and understanding the learning dynamics of PC is a long-standing problem in NN (since 2017)

**Weaknesses:**

Weakness/Cons
This paper has several weaknesses: very limited novelty,
weak related work section,
weak experimental section
results are obtained by performing sequential overfitting.

**Questions:**

# Detailed Review

* 1.	This is what authors have claimed in abstract “advancing the practical application of predictive coding theory to its full potential.” I don’t see how this is true. Let us highlight all of them in sequential order
* a.	Carefully looking into the codebase, we can observe authors are not using a validation set, which indicates sequential overfitting. Thus, results cannot be trusted
* b.	Second, the authors haven’t reported their hyper-parameters, which makes it challenging to get exact numbers.
* c.	Third if we use current codebase and change the optimizer or even learning rate, model collapses. What’s surprising to me is how adding more fc layers results in model collapse. Thus, how is this approach stable?
* d.	Now, let’s focus on results. In Table 2, we can see both PCN and SPCN get 90% accuracy on MNIST. How do such results show stability and better learning? A simple 4-layer KNN can get close to 97% accuracy, and a random network (4-layer, where only the last layer is trained) can give you close to 98%. So, how is this a stable PC? Your model is getting results that are even worse compared to random networks and KNNs.
* e.	Now let’s go to CNN results again they are worse compared to BP and also other bio-inspired approaches such as DFA, Target Prop (DTP, LRA, WM, KA) [3, 9-11], PEPITA, Sign Symmetry, and others. Even on smaller datasets, PC fails to get comparable results. A simple 8-layer MLP can give 48% on CIFAR-10, and LeNet can give close to 68% accuracy. Thus current approach in no way offer a way to use PC in practical scenarios.
* f.	All experiments are still on smaller datasets, whereas studies in bio-inspired alternatives have started showing results on larger datasets such as imagined [3, 9-11]. I would have appreciated it if the authors had worked on getting comparable performance on smaller datasets, showing stability with wider and larger networks and stability across optimizers.
* 2.	Limited Novelty and Related work section – Using skip connection to have better gradient flow or even during learning is not a new concept. Authors have failed to cite all prior works. Skip connections are used in models such as ResNet, WideResNet, DenseNet, ResNeXt, skip-rnn [6.7], and physical neural networks [5]. A few of these works have been published at this venue. Also, using skip connection is not novel; in 1948, Wiener introduced negative feedback into the control system and proposed Cybernetics. So, these approaches have been used since 1948.
* 3.	Skip connection in the backward pass. To the best of my knowledge, Sparse attentive backtracking [2] introduced skip connection at the backward pass, followed by a few recent works such as target prop variation LRA [3]. Second skip connection in predictive coding is not novel, recent survey paper from friston and his group have shown 3 variations of PC author have cited this work. However, equation 9 in this work shows a skip connection [12]. Thus, novelty seems very limited. It's surprising to see these works are neglected; even other variations of PC are neglected, which resolves several limitations mentioned in this work. The author should cite these relevant papers.
* 4.	Theoretical Results – Connecting PCN to dynamical systems is trivial. In fact, several prior works have shown the connection of NNs with dynamical systems with and without skip connections [1,4]. The connection of PCN with dynamical systems was first demonstrated by [8]. Thus, the results are natural extensions, which is nothing surprising. As one can easily show, a function is Lyapunov stable, once dynamical property holds.  Lemma-1 is widely known in linear algebra and matrix theory. Thus rather than writing as a lemma, it would be ideal to also cite books where it first appeared, such as “"Matrix Analysis" by Roger A. Horn and Charles R. Johnson”and “Introduction to Matrix Analysis" by Richard Bellman”

All missing References

1.	https://proceedings.mlr.press/v119/li20e.html (theoretical justification showing connections using skip connections and dynamical systems)

2.	https://proceedings.neurips.cc/paper_files/paper/2018/file/e16e74a63567ecb44ade5c87002bb1d9-Paper.pdf (skip connection in backward pass - SAB)

3.	https://ojs.aaai.org/index.php/AAAI/article/view/26118 (Target Prop family - LRA)

4.	https://arxiv.org/pdf/2002.05990 (few have shown skip connections leads to security weakness)

5.	https://www.nature.com/articles/s41586-021-04223-6 (Also used in physical neural networks)

6.	https://ojs.aaai.org/index.php/AAAI/article/view/4613/4491 (LSTM with dynamic skip-connection)

7.	https://openreview.net/pdf?id=HkwVAXyCW (skip-rnn)

8.	https://proceedings.mlr.press/v162/frieder22a/frieder22a.pdf (non convergence of PCNs)

9.	https://proceedings.neurips.cc/paper_files/paper/2019/file/f387624df552cea2f369918c5e1e12bc-Paper.pdf (WM and KA)

10.	https://arxiv.org/abs/2201.13415 (scaling DTP)

11. https://journals.plos.org/plosone/article?id=10.1371/journal.pone.0276427 (DTRP)

12. https://arxiv.org/pdf/2308.07870 (Friston and Rao survey paper on PC)

---

### Meta-Review · Area_Chair_XfRt · 2024-12-20

**Metareview:**

**Summary**

The paper introduces the Stable Predictive Coding Network (SPCN), which addresses the issue of instability in Predictive Coding Networks (PCNs) through innovations like length regularization and skip connection modules. The authors utilize dynamical mean-field theory to analyze the behavior of latent state lengths during training, aiming to stabilize these networks as they scale. Their proposed solution includes modifications to the training framework, such as skip connections and early stopping, to manage the norms of latent variables that tend to be disproportionally large at the end of the model and too small in intermediate layers. The effectiveness of SPCN is demonstrated through experiments on both MNIST and CIFAR-10 benchmarks, highlighting its improved learning capabilities in deeper network architectures. This work contributes to the broader fields of computational neuroscience and machine learning by enhancing the scalability of models based on local message passing.

**Strengths**

The reviewers unanimously highlighted several strengths of the proposed framework:
* The paper addresses the critical issue of training instability in Predictive Coding Networks (PCNs), which has limited their practical application despite their theoretical appeal. This instability poses a significant challenge, particularly when scaling PCNs to more complex tasks and potentially implementing them on neuromorphic hardware.
* The authors present a thorough theoretical analysis of the instability in PCNs, defining state interaction matrices and demonstrating how their proposed stabilizing techniques, including skip connections, effectively mitigate these issues. Theoretical predictions closely align with empirical results, particularly demonstrated on datasets such as MNIST.
* The paper introduces a novel training approach for PCNs that involves strategically placing skip connections between selected nodes, enhancing training efficiency by reducing the number of iterations required for the inference phase.
* The relevance of the research is underscored by its contribution to understanding biologically plausible learning mechanisms. Predictive coding is a promising candidate for such mechanisms, although it currently lacks the performance capabilities of more conventional methods like backpropagation. Improving predictive coding's effectiveness is crucial for advancing the field.

**Weaknesses**

Several core weaknesses was brought up by the reviewers. These include:
* While the introduction to Predictive Coding Networks (PCN) is clearly articulated, the overall presentation of the paper is convoluted and could significantly benefit from improved clarity and organization. For instance, some mathematical derivations and definitions are not well motivated, which makes following the paper's logic challenging.
* The lack of proper placement of the paper in the literature, motivating its importance, and novel contributions.
* The justification for focusing on the length dynamics of latent states within the framework of dynamical mean-field theory is inadequately explained.
* The paper does not adequately discuss the broader implications and interpretations of skip modules, beyond their utility in enhancing stability.
* The empirical results are inconclusive.

**Conclusion**

The majority of reviewers recognize the merits of the paper but believe that its weaknesses outweigh its strengths. While the interdisciplinary nature of the Predictive Coding Networks (PCNs) and their potential impact on machine learning are highly valued by the community, the paper fails to deliver a clear and conclusive message. Additionally, despite the authors' substantial rebuttal, it did not alter the reviewers' opinions about the paper. I agree with the reviewers and find the paper to be below the acceptance threshold. Therefore, I vote to reject this paper.

**Additional Comments On Reviewer Discussion:**

Despite my efforts to engage the reviewers in a discussion during the review period to reach a consensus on the paper’s merits and shortcomings, there was no participation in any discussion. However, given the less polarized evaluations of this paper, the discussion was not as critical for this work.

---

### Decision · Program_Chairs · 2025-01-22

Reject